# Zero-Shot Offline Imitation Learning via Optimal Transport

**Thomas Rupf** [1 2]  **Marco Bagatella** [2 3]  **Nico Gürtler** [1 2]  **Jonas Frey** [2 3]  **Georg Martius** [1 2]

## Abstract

Zero-shot imitation learning algorithms hold the promise of reproducing unseen behavior from as little as a single demonstration at test time. Existing practical approaches view the expert demonstration as a sequence of goals, enabling imitation with a high-level goal selector, and a low-level goal-conditioned policy. However, this framework can suffer from myopic behavior: the agent's immediate actions towards achieving individual goals may undermine long-term objectives. We introduce a novel method that mitigates this issue by directly optimizing the occupancy matching objective that is intrinsic to imitation learning. We propose to lift a goal-conditioned value function to a distance between occupancies, which are in turn approximated via a learned world model. The resulting method can learn from offline, suboptimal data, and is capable of non-myopic, zero-shot imitation, as we demonstrate in complex, continuous benchmarks. The code is available at https://github.com/martius-lab/zilot.

## 1. Introduction

The emergence of zero/few-shot capabilities in language modeling (Brown et al., 2020; Wei et al., 2022; Kojima et al., 2022) has renewed interest in generalist agents across all fields in machine learning. Typically, such agents are pretrained with minimal human supervision. At inference, they are capable of generalization across diverse tasks, without further training, i.e. zero-shot. Such capabilities have also been a long-standing goal in learning-based control (Duan et al., 2017). Promising results have been achieved by leveraging the scaling and generalization properties of supervised learning (Jang et al., 2022; Reed et al., 2022; O'Neill et al., 2023; Ghosh et al., 2024; Kim et al., 2024),

which however rely on large amounts of expert data, usually involving costly human participation, e.g. teleoperation. A potential solution to this issue can be found in reinforcement learning approaches, which enable learning from suboptimal data sources (Sutton & Barto, 2018). Existing methods within this framework ease the burden of learning general policies by limiting the task class to additive rewards (Laskin et al., 2021; Sancaktar et al., 2022; Frans et al., 2024) or single goals (Bagatella & Martius, 2023).

This work lifts the restrictions of previous approaches, and proposes a method that can reproduce rich behaviors from offline, suboptimal data sources. We allow arbitrary tasks to be specified through a *single* demonstration *devoid of actions* at inference time, conforming to a zero-shot Imitation Learning (IL) setting (Pathak et al., 2018; Pirotta et al., 2024). Furthermore, we consider a relaxation of this setting (Pathak et al., 2018), where the expert demonstration may be *rough*, consisting of an ordered sequence of states without precise time-step information, and *partial*, meaning each state contains only partial information about the full state. These two relaxations are desirable from a practical standpoint, as they allow a user to avoid specifying information that is either inconsequential to the task or costly to attain (e.g. only through teleoperation). For example, when tasking a robot arm with moving an object along a path, it is sufficient to provide the object's position for a few "checkpoints" without specifying the exact arm pose.

In principle, a specified goal sequence can be decomposed into multiple single-goal tasks that can be accomplished by goal-conditioned policies, as proposed by recent zero-shot IL approaches (Pathak et al., 2018; Hao et al., 2023). However, we show that this decomposition is prone to myopic behavior when expert demonstrations are partial. Continuing the robotic manipulation example from above, let us consider a task described by two sequential goals, each specifying a certain position that the object should reach. In this case an optimal goal-conditioned policy would attempt to reach the first goal as fast as possible, possibly by throwing the object towards it. The agent would then relinquish control of the object, leaving it in a suboptimal—or even unrecoverable—state. In this case, the agent would be unable to move the object towards the second goal. This myopic behavior is a fundamental issue arising from goal abstraction, as we formally argue in Section

[1]Universität Tübingen, Tübingen, Germany [2]MPI for Intelligent Systems, Tübingen, Germany [3]ETH, Zürich, Switzerland. Correspondence to: Thomas Rupf <thrupf@ethz.ch>.

*Proceedings of the 42ⁿᵈ International Conference on Machine Learning*, Vancouver, Canada. PMLR 267, 2025. Copyright 2025 by the author(s).

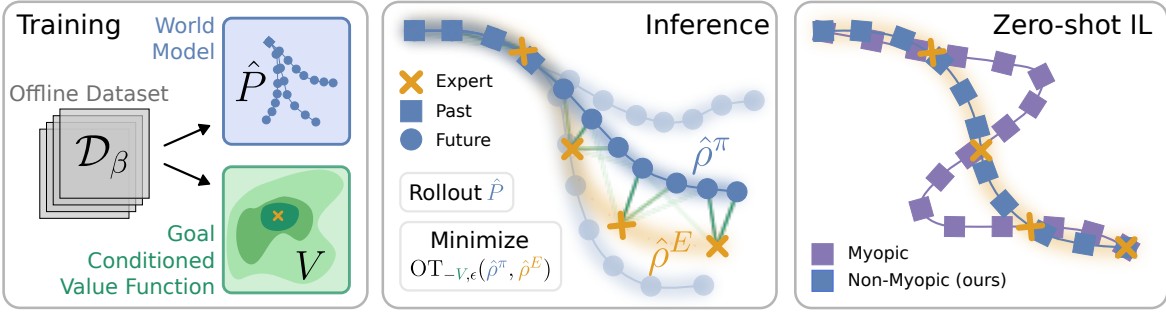

*Figure 1.* Overview of ZILOT. After learning a world model $\hat{P}$ and a goal-conditioned value function $V$ from offline data (left), a zero-order optimizer directly matches the occupancy of rollouts $\hat{\rho}^\pi$ from the learned world model to the occupancy of a single expert demonstration $\hat{\rho}^E$ (center). This is done by lifting the goal-conditioned value function to a distance between occupancies using Optimal Transport. The resulting policy displays non-myopic behavior (right).

3, and results in catastrophic failures in hard-to-control environments, as we demonstrate empirically in Section 5.

In this work we instead provide a holistic solution to zero-shot offline imitation learning by adopting an occupancy matching formulation. We name our method ZILOT (**Z**ero-shot Offline **I**mitation **L**earning from **O**ptimal **T**ransport). We utilize Optimal Transport (OT) to lift the state-goal distance inherent to GC-RL to a distance between the expert's and the policy's occupancies, where the latter is approximated by querying a learned world model. Furthermore, we operationalize this distance as an objective in a standard fixed horizon MPC setting. Minimizing this distance leads to non-myopic behavior in zero-shot imitation. We verify our claims empirically by comparing our planner to previous zero-shot IL approaches across multiple robotic simulation environments, down-stream tasks, and offline datasets.

## 2. Preliminaries

### 2.1. Imitation Learning

We model an environment as a controllable Markov Chain[1] $\mathcal{M} = (\mathcal{S}, \mathcal{A}, P, \mu_0)$, where $\mathcal{S}$ and $\mathcal{A}$ are state and action spaces, $P : \mathcal{S} \times \mathcal{A} \to \Omega(\mathcal{S})$[2] is the transition function and $\mu_0 \in \Omega(\mathcal{S})$ is the initial state distribution. In order to allow for partial demonstrations, we additionally define a goal space $\mathcal{G}$ and a surjective function $\phi : \mathcal{S} \to \mathcal{G}$ which maps each state to its abstract representation. To define "goal achievement", we assume the existence of a goal metric $h$ on $\mathcal{G}$ that does not need to be known. We then regard state $s \in \mathcal{S}$ as having achieved goal $g \in \mathcal{G}$ if we have $h(\phi(s), g) < \epsilon$ for some fixed $\epsilon > 0$. For each policy $\pi : \mathcal{S} \to \Omega(\mathcal{A})$, we can measure the (undiscounted) $N$-step state and goal occupancies respectively as

$$\varrho_N^\pi(s) = \frac{1}{N+1} \sum_{t=0}^{N} \Pr[s = s_t] \tag{1}$$

[1]or reward-free Markov Decision Process.
[2]where $\Omega(\mathcal{S})$ denotes the set of distributions over $\mathcal{S}$.

and

$$\rho_N^\pi(g) = \frac{1}{N+1} \sum_{t=0}^{N} \Pr[g = \phi(s_t)], \tag{2}$$

where $s_0 \sim \mu_0$, $s_{t+1} \sim P(s_t, a_t)$ and $a_t \sim \pi(s_t)$. These quantities are particularly important in the context of imitation learning. We refer the reader to (Liu et al., 2023) for a full overview over IL settings, and limit this discussion to offline IL with rough and partial expert trajectories. It assumes access to two datasets: $\mathcal{D}_\beta = (s_0^i, a_0^i, s_1^i, a_1^i, \dots)_{i=1}^{|\mathcal{D}_\beta|}$ consisting of full state-action trajectories from $\mathcal{M}$ and $\mathcal{D}_E = (g_0^i, g_1^i, \dots)_{i=1}^{|\mathcal{D}_E|}$ containing demonstrations of an expert in the form of goal sequences, not necessarily abiding to the dynamic of $\mathcal{M}$. Note that both datasets do not have reward labels. The goal is to train a policy $\pi$ that imitates the expert, which is commonly formulated as matching goal occupancies

$$\rho_N^\pi \overset{D}{=} \rho_N^{\pi_E}. \tag{3}$$

The setting we consider in this work is *zero-shot* offline IL which imposes two additional constraints on offline IL. First, $\mathcal{D}_E$ is only available at inference time, which means pre-training has to be task-agnostic. We further assume $\mathcal{D}_E$ consists of a single trajectory $(g_0, \dots, g_m) = g_{0:m}$. Second, at inference, the agent should imitate $\pi_E$, with a "modest compute-overhead" (Pathak et al., 2018; Touati & Ollivier, 2021; Pirotta et al., 2024). In practice, imitation of unseen trajectories should be order of magnitudes cheaper than IL from scratch, and largely avoid costly operations (e.g. network updates).

### 2.2. Optimal Transport

In the field of machine learning, it is often of interest to match distributions, i.e. find some probability measure $\mu$ that resembles some other probability measure $\nu$. In recent years there has been an increased interest in Optimal Transportation (OT) (Amos et al., 2023; Haldar et al., 2022;

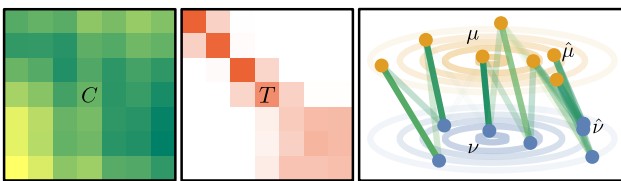

*Figure 2.* An example of Optimal Transport between the discrete approximation $\hat{\mu}, \hat{\nu}$ of two Gaussians $\mu, \nu$. The cost matrix $C$ consists of the point-wise costs where the cost here is the Euclidian distance. A coupling matrix $T \in \mathcal{U}(\hat{\mu}, \hat{\nu})$ (middle) is visualized through lines representing the matching (right).

Bunne et al., 2023; Pooladian et al., 2024). As illustrated in figure 2, OT does not only compare probability measures in a point-wise fashion, like $f$-Divergences such as the Kullback-Leibler Divergence ($D_{\mathrm{KL}}$), but also incorporates the geometry of the underlying space. This also makes OT robust to empirical approximation (sampling) of probability measures ((Peyré & Cuturi, 2019), p.129). Formally, OT describes the coupling $\gamma \in \mathcal{P}(\mathcal{X} \times \mathcal{Y})$ of two measures $\mu \in \mathcal{P}(\mathcal{X}), \nu \in \mathcal{P}(\mathcal{Y})$ with minimal transportation cost w.r.t. some cost function $c : \mathcal{X} \times \mathcal{Y} \to \mathbb{R}$. The primal Kantorovich form is given as the optimization problem

$$\mathrm{OT}_c(\mu, \nu) = \inf_{\gamma \in \mathcal{U}(\mu, \nu)} \int_{\mathcal{X} \times \mathcal{Y}} c(x_1, x_2) d\gamma(x_1, x_2) \quad (4)$$

where the optimization is over all joint distributions of $\mu$ and $\nu$ denoted as $\gamma \in \mathcal{U}(\mu, \nu)$ (*couplings*). If $\mathcal{X} = \mathcal{Y}$ and $(\mathcal{X}, c)$ is a metric space then for $p \in \mathbb{N}$, $W_p^p = \mathrm{OT}_{c^p}$ is called the Wasserstein-$p$ distance which was shown to be a metric on the subset of measures on $\mathcal{X}$ with finite $p$-th moments (Clement & Desch, 2008).

Given samples $x_1, \ldots, x_n \sim \mu$ and $y_1, \ldots, y_m \sim \nu$ the discrete OT problem between the discrete probability measures $\hat{\mu} = \sum_{i=1}^{n} a_i \delta_{x_i}$ and $\hat{\nu} = \sum_{j=1}^{m} b_j \delta_{y_j}$ can be written as a discrete version of equation 4, namely

$$\mathrm{OT}_c(\hat{\mu}, \hat{\nu}) = \min_{\boldsymbol{T} \in \mathcal{U}(\boldsymbol{a}, \boldsymbol{b})} \sum_{i=1}^{n} \sum_{j=1}^{m} c(x_i, y_j) T_{ij} \quad (5)$$

$$= \min_{\boldsymbol{T} \in \mathcal{U}(\boldsymbol{a}, \boldsymbol{b})} \langle \boldsymbol{C}, \boldsymbol{T} \rangle \quad (6)$$

with the cost matrix $C_{ij} = c(x_i, y_j)$. The marginal constraints can now be written as $\mathcal{U}(\boldsymbol{a}, \boldsymbol{b}) = \{\boldsymbol{T} \in \mathbb{R}^{n \times m} : \boldsymbol{T} \cdot \boldsymbol{1}_m = \boldsymbol{a} \text{ and } \boldsymbol{T}^\top \cdot \boldsymbol{1}_n = \boldsymbol{b}\}$. This optimization problem can be solved via Linear Programming. Furthermore, Cuturi (2013) shows that the entropically regularized version, commonly given as $\mathrm{OT}_{c,\eta}(\hat{\mu}, \hat{\nu}) = \min_{\boldsymbol{T} \in \mathcal{U}(\boldsymbol{a}, \boldsymbol{b})} \langle \boldsymbol{C}, \boldsymbol{T} \rangle - \eta D_{\mathrm{KL}}(\boldsymbol{T}, \boldsymbol{a}\boldsymbol{b}^\top)$, can be efficiently solved in its dual form using Sinkhorn's algorithm (Sinkhorn & Knopp, 1967).

### 2.3. Goal-conditioned Reinforcement Learning

As techniques from the literature will be recurring in this work, we provide a short introduction to fundamental ideas

in GC-RL. We can introduce this framework by enriching the controllable Markov Chain $\mathcal{M}$. We condition it on a goal $g \in \mathcal{G}$ and cast it as an (undiscounted) Markov Decision Process $\mathcal{M}_g = (\mathcal{S} \cup \{\bot\}, \mathcal{A}, P_g, \mu_0, R_g, T_{\max})$. Compared to the reward-free setting above, the dynamics now include a sink-state $\bot$ upon goal-reaching and a reward of $-1$ until this happens:

$$P_g(s, a) = \begin{cases} P(s, a) & \text{if } h(\phi(s), g) \geq \epsilon \\ \delta_\bot & \text{otherwise} \end{cases} \quad (7)$$

$$R_g(s, a) = \begin{cases} -1 & \text{if } h(\phi(s), g) \geq \epsilon \\ 0 & \text{otherwise} \end{cases} \quad (8)$$

where $\delta_x$ stands for the probability distribution assigning all probability mass to $x$.

We can now define the goal-conditioned value function as

$$V^\pi(s_0, g) = \mathop{\mathbb{E}}_{\mu_0, P_g, \pi} \left[ \sum_{t=0}^{T_{\max}} R_g(s_t, a_t) \right] \quad (9)$$

where $s_0 \sim \mu_0, s_{t+1} \sim P_g(s_t, a_t), a_t \sim \pi(s_t, g)$. The optimal goal-conditioned policy is then $\pi^\star = \arg\max_\pi \mathbb{E}_{g \sim \mu_{\mathcal{G}}, s \sim \mu_0} V^\pi(s_0; g)$ for some goal distribution $\mu_{\mathcal{G}} \in \Omega(\mathcal{G})$. Intuitively, the value function $V^\pi(s, g)$ corresponds to the negative number of expected steps that $\pi$ needs to move from state $s$ to goal $g$. Thus the distance $d = -V^\star$ corresponds to the expected first hit time. If no goal abstraction is present, i.e. $\phi = \mathrm{id}_\mathcal{S}$, then $(\mathcal{S}, d)$ is a quasimetric space (Wang et al., 2023), i.e. $d$ is non-negative and satisfies the triangle inequality. Note, though, that $d$ does not need be be symmetric.

## 3. Goal Abstraction and Myopic Planning

The distribution matching objective at the core of IL problems is in general hard to optimize. For this reason, traditional methods for zero-shot IL leverage a hierarchical decomposition into a sequence of GC-RL problems (Pathak et al., 2018; Hao et al., 2023). We will first describe this approach, and then show how it potentially introduces myopic behavior and suboptimality.

In the pretraining phase, Pathak et al. (2018) propose to train a goal-conditioned policy $\pi_g : \mathcal{S} \times \mathcal{G} \to \mathcal{A}$ on reaching single goals and a goal-recognizer $C : \mathcal{S} \times \mathcal{G} \to \{0, 1\}$ that detects whether a given state achieves the given goal. Given an expert demonstration $g_{1:M}$ and an initial state $s_0$, imitating the expert can then be sequentially decomposed into $M$ goal-reaching problems, and solved with a hierarchical agent consisting of two policies. On the lower level, $\pi_g$ chooses actions to reach the current goal; on the higher level, $C$ decides whether the current goal is achieved and $\pi_g$ should target the next goal in the sequence.

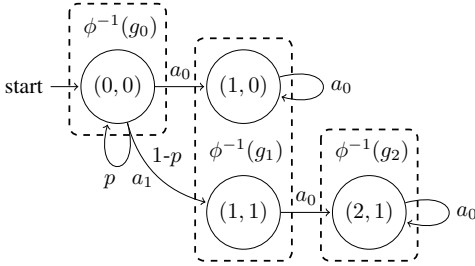

*Figure 3.* Controllable Markov Chain with $\phi : (x, y) \mapsto x$.

We define the pre-image $\phi^{-1}(g) = \{s \in \mathcal{S} : \phi(s) = g\}$ as the set of all states that map to a goal, and formalize the suboptimality of the above method under goal abstraction as follows.

**Proposition 3.1.** *Let us define the optimal classifier* $C(s, g) = \mathbf{1}_{h(\phi(s), g) < \epsilon}$. *Given a set of visited states* $\mathcal{P} \subseteq \mathcal{S}$, *the current state* $s \in \mathcal{P}$, *and a goal sequence* $g_{1:M} \in \mathcal{G}^M$, *let the optimal hierarchical policy be* $\pi_h^\star(s) = \pi^\star(s, g_{i+1})$, *where* $i$ *is the smallest integer such that there exist a state* $s_p \in \mathcal{P}$ *with* $h(\phi(s_p), g_i) < \epsilon$, *and* $i = 0$ *otherwise. There exists a controllable Markov Chain* $\mathcal{M}$ *and a realizable sequence of goals* $g_{0:M}$ *such that, under a suitable goal abstraction* $\phi(\cdot)$, $\pi_h^\star$ *will not reach all goals in the sequence, i.e.* $\rho_N^{\pi_h^\star}(g_i) = 0$ *for some* $i \in [0, \ldots, M]$ *and all* $N \in \mathbb{N}$.

*Proof.* Consider the Markov Chain $\mathcal{M}$ depicted in figure 3 with goal abstraction $\phi : (x, y) \mapsto x$ and $p > 0$. Now, consider the goal sequence $(g_0, g_1, g_2) = (0, 1, 2)$, which can only be achieved, by a policy taking action $a_1$ in the initial state $s_0 = (0, 0)$. Consider $\pi_h^\star$ in $s_0$, with $\mathcal{P} = \{s_0\}$. The smallest integer $i$ such that $h(\phi(s_0), g_i) < \epsilon$ is $i = 0$, therefore $\pi_h^\star(s_0) = \pi^\star(s_0, g_1)$. We can then compare the state-action values $Q$ in $s_0$:

$$Q^{\pi^\star(\cdot, g_1)}(s_0, a_1, g_1) = \sum_{t=0}^{T_{\max}} -p^t = -1 \cdot \frac{1 - p^{(T_{\max}+1)}}{1 - p}$$
$$< -1 = Q^{\pi^\star(\cdot, g_1)}(s_0, a_0, g_1). \quad (10)$$

This implies that $\pi_h^\star(s_0) = \pi^\star(s_0, 1) = a_0$. The next state visited by $\pi_h^\star$ will always be $(1, 0)$, from which $(2, 1)$ is not reachable, and $g_2$ is not achievable. We thus have $\rho_N^{\pi_h^\star}(g_2) = 0$ for all $N \in \mathbb{N}$. $\square$

We remark that this issue arises in the presence of goal abstraction which plays a vital role in the partial demonstration setting we consider. Without goal abstraction, i.e., if each goal is fully specified, there is no leeway in how to achieve it for the policy (assuming $\epsilon \to 0$ as well). Nevertheless, goal abstraction is ubiquitous in practice (Schaul et al., 2015) and necessary to enable learning in complex environments (Andrychowicz et al., 2017).

## 4. Optimal Transport for Zero-Shot IL

Armed with recent tools in value estimation, model-based RL and trajectory optimization, we propose a method for zero-shot offline imitation learning that *directly* optimizes the occupancy matching objective, introducing only minimal approximations. As a result, the degree of myopia is greatly reduced, as we show empirically in section 5.

In particular, we propose to solve the occupancy matching problem in equation 3 by minimizing the Wasserstein-1 metric $W_1$ with respect to goal metric $h$ on the goal space $\mathcal{G}$, i.e.

$$W_1(\rho_N^\pi, \rho_N^E) = \text{OT}_h(\rho_N^\pi, \rho_N^E). \quad (11)$$

This objective involves two inaccessible quantities: goal occupancies $\rho_N^\pi, \rho_N^E$, as well as the goal metric $h$. Our key contribution lies in how these quantities can be practically estimated, enabling optimization of the objective with scalable deep RL techniques.

**Occupancy Estimation** Since the expert's and the policy's occupancy are both inaccessible, we opt for discrete, sample-based approximations. In the case of the expert occupancy $\rho_N^E$, the single trajectory provided at inference $(g_0, \ldots, g_M)$ represents a valid sample from it, and we use it directly. For an arbitrary agent policy $\pi$, we use a discrete approximation after training a dynamics model $\hat{P} \approx P$ on $\mathcal{D}_\beta$, which can be done offline through standard supervised learning. We can then approximate $\rho_N^\pi$ by jointly rolling out the learned dynamics model and the policy $\pi$. We thus get the discrete approximations

$$\rho_N^E \approx \hat{\rho}_M^E = \frac{1}{M + 1} \sum_{j=0}^{M} \delta_{g_j} \quad \text{and} \quad (12)$$

$$\rho_N^\pi \approx \hat{\rho}_N^\pi = \frac{1}{N + 1} \sum_{t=0}^{N} \delta_{\phi(s_t)} \quad (13)$$

where for the latter we sample

$$s_0 \sim \mu_0, s_{t+1} \sim \hat{P}(s_t, a_t), a_t \sim \pi(s_t). \quad (14)$$

Similarly, we can also obtain an estimate for the *state* occupancy of $\pi$ as $\varrho_N^\pi \approx \hat{\varrho}_N^\pi = \frac{1}{N+1} \sum_{t=0}^{N} \delta_{s_t}$.

**Metric Approximation** As $h$ may be unavailable or hard to specify in practical settings, we propose to train a goal-conditioned value function $V^\star$ from the offline data $\mathcal{D}_\beta$ and use the distance $d(s, g) = -V^\star(s, g)$ (i.e. the learned first hit time) as a proxy. For a given state-goal pair $(s, g)$, this corresponds to the approximation $d(s, g) \approx h(\phi(s), g)$. It is easy to show that a minimizer of $h(\phi(\cdot), g)$ also minimizes $d(\cdot, g)$. Using $d$ also has the benefit of incorporating the dynamics of the MDP into the cost of the OT problem. The

use of this distance has seen some use as the cost function in Wasserstein metrics between state occupancies in the past (Durugkar et al., 2021). As we show in section 5.3, $d$ is able to capture potential asymmetries in the MDP, while remaining informative of $h$. We note that, while $h : \mathcal{G} \times \mathcal{G} \to \mathbb{R}$ is a distance in goal-space, $d : \mathcal{S} \times \mathcal{G} \to \mathbb{R}$ is a distance between states and goals. Nonetheless, $d$ remains applicable as the policy's occupancy can also be estimated in state spaces as $\hat{\varrho}_N^\pi$. Given the above considerations, we can rewrite our objective as the discrete optimal transport problem

$$\pi^\star = \arg\min_\pi \text{OT}_d(\hat{\varrho}_N^\pi, \hat{\rho}_M^E). \tag{15}$$

**Optimization** Having addressed density and metric approximations, we now focus on optimizing the objective in equation 15. Fortunately, as a discrete OT problem, the objective can be evaluated efficiently using Sinkhorn's algorithm when introducing entropic regularization with a factor $\eta$ (Cuturi, 2013; Peyré & Cuturi, 2019). A non-Markovian, deterministic policy optimizing the objective at state $s_k \in \mathcal{S}$ can be written as

$$\pi(s_{0:k}, g_{0:m}) \tag{16}$$
$$\approx \arg\min_{a_k} \min_{a_{k+1:N-1}} \text{OT}_{d,\eta} \left( \frac{1}{N+1} \sum_{i=0}^N \delta_{s_i}, \frac{1}{M+1} \sum_{j=0}^M \delta_{g_j} \right)$$

where $s_{0:k}$ are the states visited so far and $s_{k+1:N}$ are rolled out using the learned dynamics model $\hat{P}$ and actions $a_{k:N-1}$. Note that while $s_{0:k}$ are part of the objective, they are constant and are not actively optimized.

Intuitively, this optimization problem corresponds to finding the first action from a sequence $(a_{k:N-1})$ that minimizes the OT costs between the empirical expert goal occupancy, and the induced empirical policy state occupancy. This type of optimization problem fits naturally into the framework of planning with zero-order optimizers and learned world models (Chua et al., 2018; Ha & Schmidhuber, 2018); while these algorithms are traditionally used for additive costs, the flexibility of zero-order optimizers (Rubinstein & Kroese, 2004; Williams et al., 2015; Pinneri et al., 2020) allows a straightforward application to our problem. The objective in equation 17 can thus be directly optimized with CEM variants (Pinneri et al., 2020) or MPPI (Williams et al., 2015), in a model predictive control (MPC) fashion.

Like for other MPC approaches, we are forced to plan for a finite horizon $H$, which might be smaller than $N$, because of imperfections in the learned dynamics model or computational constraints. This is referred to as receding horizon control (Datko, 1969). When the policy rollouts used for computing $\hat{\varrho}_N^\pi$ are truncated, it is also necessary to truncate the goal sequence to exclude any goals that cannot be

reached within $H$ steps. To this end, we train an extra value function $W$ that estimates the number of steps required to go from one goal to the next by regressing onto $V$, i.e. by minimizing $\mathbb{E}_{s,s'\sim\mathcal{D}_\beta}[(W(\phi(s); \phi(s')) - V(s; \phi(s')))^2]$. For $i \in [0, \ldots, M]$, we can then estimate the time when $g_i$ should be reached as

$$t_i \approx -V(s_0; g_0) - \sum_{j=1}^i W(g_{j-1}; g_j). \tag{17}$$

We then simply truncate the online problem to only consider goals relevant to $s_1, \ldots, s_{k+H}$, i.e. $g_0, \ldots, g_K$ where $K = \min\{j : t_j \geq k+H\}$. We note that this approximation of the infinite horizon objective can potentially result in myopic behavior if $K < M$; nonetheless, optimal behavior is recovered as the effective planning horizon increases. Algorithm 1 shows how the practical OT objective is computed.

---

**Algorithm 1** OT cost computation for ZILOT

---

**Require:** Pretrained GC value functions $V, W$ and dynamics model $\hat{P}$; horizon $H$, solver iterations $r$ and regularization factor $\eta$.

**Initialization:** State $s_0$ and expert trajectory $g_{1:M}$, precomputed $t_{0:M}$

**input** State history $s_{0:k}$, future actions $a_{k:k+H-1}$
    {Rollout learned dynamics}
  $s_{k+1:k+H} \leftarrow \texttt{rollout}(\hat{P}, s_k, a_{k:k+H-1})$
    {Compute which goals are reachable}
  $K \leftarrow \min\{j : t_j \geq k+H\}$
    {Compute cost matrix}
  $C_{ij} \leftarrow -V(s_i; g_j), i \in \{0, \ldots, k+H\}, j \in \{0, \ldots, K\}$
    {Compute uniform marginals}
  $\boldsymbol{a} \leftarrow \frac{1}{k+H+1}\mathbf{1}_{k+H+1}, \boldsymbol{b} \leftarrow \frac{1}{K+1}\mathbf{1}_{K+1}$
    {Run Sinkhorn Algorithm}
  $\boldsymbol{T} \leftarrow \texttt{sinkhorn}(\boldsymbol{a}, \boldsymbol{b}, \boldsymbol{C}, r, \eta)$
**output** $\sum_{ij} T_{ij}C_{ij}$ {Return OT cost}

---

**Implementation** The method presented relies solely on three learned components: a dynamics model $\hat{P}$, and the state-goal and goal-goal GC value functions $V$ and $W$. All of them can be learned offline from the dataset $\mathcal{D}_\beta$. In practice, we found that several existing deep reinforcement learning frameworks can be easily adapted to learn these functions. We adopt TD-MPC2 (Hansen et al., 2024), a state of the art model-based algorithm that has shown promising results in single- and multitask online and offline RL. We note that planning takes place in the latent space constructed by TD-MPC2's encoders. We adapt the method to allow estimation of goal-conditioned value functions, as described in appendix C. We follow prior work (Andrychowicz et al., 2017; Bagatella & Martius, 2023; Tian et al., 2021) and sample goals from the future part of trajectories in $\mathcal{D}_\beta$ in order to synthesize rewards without supervision. We note

that this goal-sampling method also does not require any knowledge of $h$.

## 5. Experiments

This section constitutes an extensive empirical evaluation of ZILOT for zero-shot IL. We first describe our experimental settings, and then present qualitative and quantitative result, as well as an ablation study. We consider a selection of 30 tasks defined over 5 environments, as summarized below and described in detail in appendix A and C.

`fetch` (Plappert et al., 2018) is a manipulation suite in which a robot arm either pushes (Push), or lifts (Pick&Place) a cube towards a goal. To illustrate the failure cases of myopic planning, we also evaluate a variation of Push (i.e. Slide), in which the table size exceeds the arm's range, the table's friction is reduced, and the arm is constrained to be touching the table. As a result, the agent cannot fully constrain the cube, e.g. by picking it up, or pressing on it, and the environment strongly punishes careless manipulation. In all three environments, tasks consist of moving the cube along trajectories shaped like the letters "S", "L", and "U".

`halfcheetah` (Wawrzyński, 2009) is a classic Mujoco environment where the agent controls a cat-like agent in a 2D horizontal plane. As this environment is not goal-conditioned by default, we choose the x-coordinate and the orientation of the cheetah as a meaninful goal-abstraction. This allows the definition of tasks involving standing up and hopping on front or back legs, as well as doing flips.

`pointmaze` (Fu et al., 2021) involves maneuvering a point-mass through a maze via force control. Downstream tasks consist of following a series of waypoints through the maze.

**Planners** The most natural comparison is the framework proposed by Pathak et al. (2018), which addresses imitation through a hierarchical decomposition, as discussed in section 3. Both hierarchical components are learned within TD-MPC2: the low-level goal-conditioned policy is by default part of TD-MPC2, while the goal-classifier (Cls) can be obtained by thresholding the learned value function $V$. We privilege this baseline (**Policy+Cls**) by selecting the threshold minimizing $W_{\min}$ *per environment* among the values $\{1, 2, 3, 4, 5\}$. Moreover, we also compare to a version of this baseline replacing the low-level policy with zero-order optimization of the goal-conditioned value function (**MPC+Cls**), thus ablating any benefits resulting from model-based components. We remark that all MPC methods use the same zero-order optimizer iCEM (Pinneri et al., 2020). We further compare ZILOT to $\mathbf{ER_{FB}}$ and $\mathbf{RER_{FB}}$, two approaches that combine zero-shot RL and reward-based IL using the Forward-Backward (FB) framework (Pirotta et al., 2024). We refer the reader to

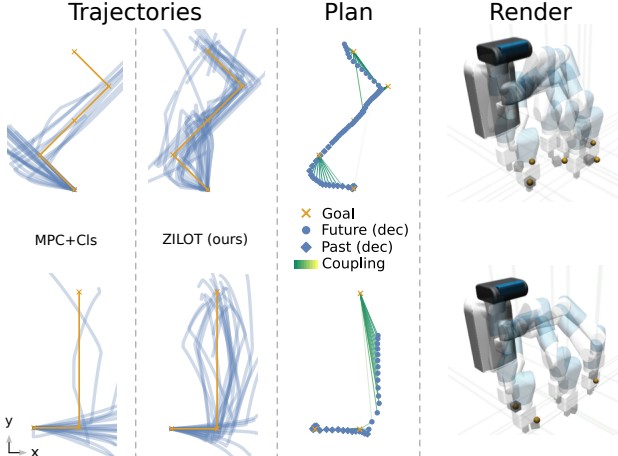

Figure 4. Example tasks in `fetch_slide_large_2D`. The left two columns show five trajectories across five seeds of the myopic method MPC+Cls and ZILOT (ours). The trajectories are drawn in the $x$-$y$-plane of the goal space and just show the movement of the cube. ZILOT's behavior imitates the given goal trajectories more closely. On the right, we visualize the OT objective at around three quarters of the episode time. It includes both the past and planned future states, as well as their coupling to the goals. Note that planning occurs in the latent state of TD-MPC2, and separately trained decoders are used for this visualization.

appendix B for a discussion of all FB-IL approaches in our rough and partial setting.

**Metrics** We report two metrics for evaluating planner performance. The first one is the minimal encountered (empirical) Wasserstein-1 distance under the goal metric $h$ of the agent's trajectory and the given goal sequence. Formally, given trajectory $(s_0, \dots, s_N)$ and the goal sequence $(g_0, \dots, g_M)$ we define $W_{\min}(s_{0:N}, g_{1:M})$ as

$$\min_{k \in \{0,\dots,N\}} W_1\left(\frac{1}{k+1}\sum_{i=0}^{k}\delta_{\phi(s_i)}, \frac{1}{M+1}\sum_{j=0}^{M}\delta_{g_j}\right). \quad (18)$$

We introduce a secondary metric "GoalFraction", which represents the fraction of goals that are achieved in the order they were given. Formally, this corresponds to the length of the longest subsequence of achieved goals that matches the desired order.

### 5.1. Can ZILOT effectively imitate unseen trajectories?

We first set out to qualitatively evaluate whether the method is capable of imitation in complex environments, despite practical approximations. Figure 4 illustrates how MPC+Cls and ZILOT imitate an expert sliding a cube across the big table of the `fetch_slide_large_2D` environment. The myopic baseline struggles to regain control over the cube after moving it towards the second goal, leading to trajectories that leave the manipulation range. In contrast, ZILOT

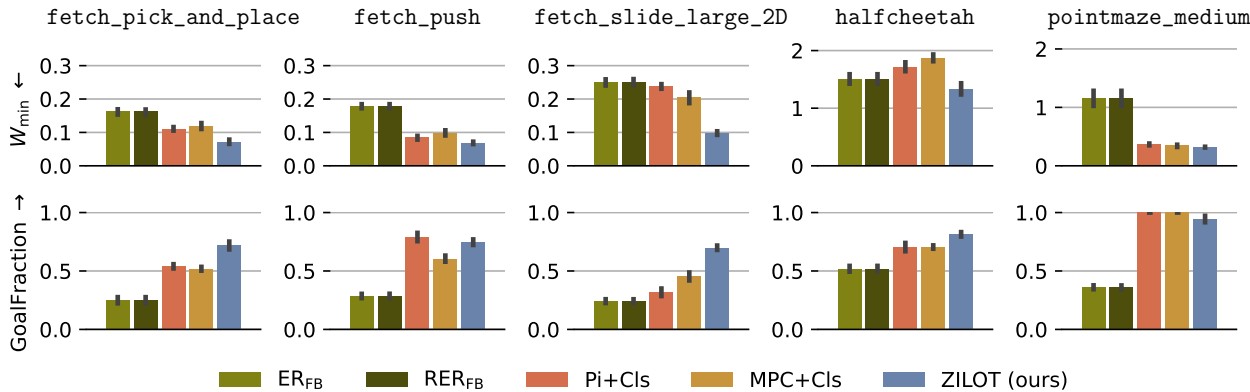

*Figure 5.* Performance comparison of ZILOT and other methods aggregated over environments. Table 1 reports more detailed results.

plans beyond the second goal. As displayed in the middle part of figure 4, the coupling of the OT problem approximately pairs up each state in the planned trajectory with the appropriate goal, leading to closer imitation of the expert.

### 5.2. How does ZILOT perform compared to prior methods?

We provide a quantitative evaluation of ZILOT with respect to the other planners in table 5. For more details we refer the reader to appendix A. As ZILOT directly optimizes a distribution matching objective, it generally reproduces expert trajectories more closely, achieving a lower Wasserstein distance to its distribution. This is especially evident in environments that are very punishing to myopic planning, such as the Fetch Slide environment shown in figure 4. In most environments, our method also out-performs the baselines in terms of the fraction of goals reached. In less punishing environments, ZILOT may sacrifice precision in achieving the next goal exactly for an overall closer match of the expert trajectory which is most clearly visible in the `pointmaze` environment. We note that the performance of the two myopic baselines Pi+Cls and MPC+Cls are very similar, suggesting that the performance gap to ZILOT stems from the change in objective, rather than implementation or model-based components. We suspect the origins of the subpar performance of $ER_{FB}$ and $RER_{FB}$ are two-fold. First, the FB framework (Touati & Ollivier, 2021; Pirotta et al., 2024) has been found to underperform in low-data regimes(Jeen et al., 2024). Second, $ER_{FB}$ and $RER_{FB}$ use a regularized f-divergence objective, which translates to an RL problem with additive rewards. As (Pirotta et al., 2024) state, this regularization comes at a cost, particularly if states do not contain dynamical information or in ergodic MDPs. In this case, a policy can optimize the reward by remaining in the most likely expert state, yielding a degenerate solution. Conversely, such a solution would be discarded by ZILOT as it uses an unregularized objective.

### 5.3. What matters for ZILOT?

To validate some of our design choices we finally evaluate the following versions of our method.

- **OT+unbalanced**, our method with unbalanced OT (Liero et al., 2018; Séjourné et al., 2019), which turns the hard marginal constraint $\mathcal{U}$ (see section 2.2) into a soft constraint. We use this method to address the fact that a rough expert trajectory may not necessarily yield a feasible expert occupancy approximation.
- **OT+Cls**, a version of our method which uses the classifier (Cls) (with the same hyperparameter search) to discards all goals that are recognized as reached. This allows this method to only consider future goals and states in the OT objective.
- **OT+$h$**, our method with the goal metric $h$ on $\mathcal{G}$ as the cost function in the OT problem, replacing $d$.

Our results are summarized in figure 6. First, we see that using unbalanced OT does not yield significant improvements. Second, using a goal-classifier can have a bad impact on matching performance. We suspect this is the case because keeping track of the history of states gives a better, more informative, estimate of which part of the expert occupancy has already been fulfilled. Finally, we observe that the goal metric $h$ may not be preferable to $d$, even if it is available. We mainly attribute this to the fact that, in the considered environments, any action directly changes the state occupancy, but the same cannot be said for the goal occupancy. Since $h$ only allows for the comparison of goal occupancies, the optimization landscape can be very flat in situations where most actions do not change the future state trajectory under goal abstraction, such as the start of `fetch` tasks as visible in its achieved trajectories in the figures in appendix E. Furthermore, while $h$ is locally accurate, it ignores the global geometry of MDPs, as shown by its poor performance in strongly asymmetric environments (i.e., `halfcheetah`).

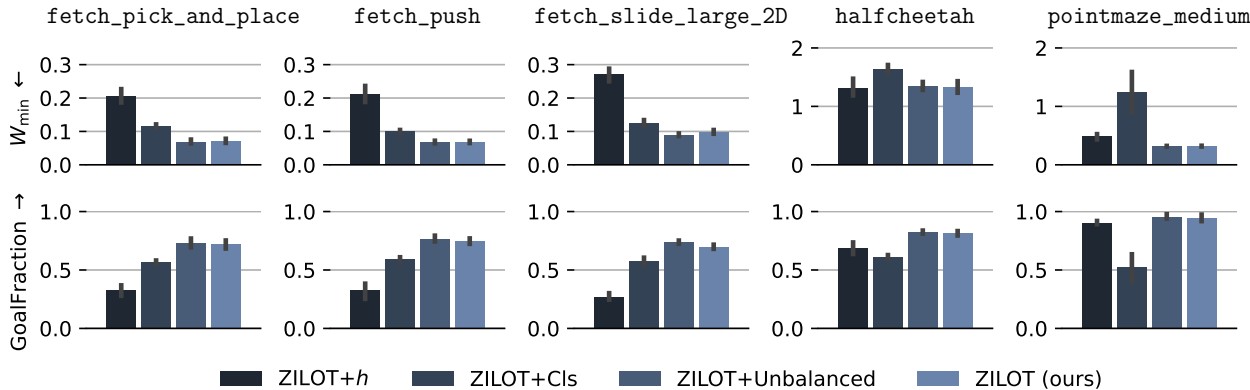

*Figure 6.* Ablation of design choices in ZILOT, including coupling constraints (OT+unbalanced), partial trajectory matching (OT+Cls), and the approximation of $h$ by $d$ (OT+$h$). For more detailed results, please refer to table 2.

## 6. Related Work

**Zero-shot IL** When a substantial amount of compute is allowed at inference time, several existing methods leverage pretrained models to infer actions, and retrieve an imitator policy via behavior cloning (Pan et al., 2020; Zhang et al., 2023; Torabi et al., 2018). As already discussed in section 3, most (truly) zero-shot methods cast the problem of imitating an expert demonstration as following the sequence of its observations (Pathak et al., 2018; Hao et al., 2023). Expert demonstrations are then imitated by going from one goal to the next using a goal-conditioned policy. In contrast, our work proposes a holistic approach to imitation, which considers all goals within the planning horizon.

**Zero-Shot RL** Vast amounts of effort have been dedicated to learning generalist agents without supervision, both on the theoretical (Touati & Ollivier, 2021; Touati et al., 2023) and practical side (Laskin et al., 2021; Mendonca et al., 2021). Among others, (Sancaktar et al., 2022; P. et al., 2021; Bagatella & Martius, 2023) learn a dynamics model through curious exploration and show how it can be leveraged to optimize additive objectives. More recently, Frans et al. (2024) use Functional Reward Encodings to encode arbitrary additive reward functions in a latent that is used to condition a policy. While these approaches are effective in a standard RL setting, they are not suitable to solve instances of global RL problems (Santi et al., 2024) (i.e., distribution matching). One notable exception is the forward-backward framework (Touati & Ollivier, 2021; Pirotta et al., 2024), which we discuss in detail in appendix B.

**Imitation Learning** A range of recent work has been focused on training agents that imitate experts from their trajectories by matching state, state-action, or state-next-state occupancies depending on what is available. These methods either directly optimize various distribution matching objectives (Liu et al., 2023; Ma et al., 2022) or recover a reward using Generative Adversarial Networks (GAN)

(Ho & Ermon, 2016; Li et al., 2023) or in one instance OT (Luo et al., 2023). Another line of work has shown impressive real-world results by matching the action distributions (Shafiullah et al., 2022; Florence et al., 2021; Chi et al., 2023) directly. All these approaches do not operate in a zero-shot fashion, or need ad-hoc data collection.

**OT in RL** Previous works have often used OT as a reward signal in RL. One application is online fine-tuning, where a policy's rollouts are rewarded in proportion to how closely they match expert trajectories (Dadashi et al., 2021; Haldar et al., 2022). Luo et al. (2023) instead use a similar trajectory matching strategy to recover reward labels for unlabelled mixed-quality offline datasets. Most of the works mentioned above rely on simple Cosine similarities and Euclidean distances as cost-functions their OT problems..

## 7. Discussion

In this work, we point out a failure-mode of current zero-shot IL methods that cast imitating an expert demonstration as following a sequence of goals with myopic GC-RL policies. We address this issue by framing the problem as occupancy matching. By introducing discretizations and minimal approximations, we derive an Optimal Transportation problem that can be directly optimized at inference time using a learned dynamics model, goal-conditioned value functions, and zero-order optimizer. Our experimental results across various environments and tasks show that our approach outperforms state-of-the-art zero-shot IL methods, particularly in scenarios where non-myopic planning is crucial. We additionally validate our design choices through a series of ablations.

From a practical standpoint, our method is mainly limited in its reliance on a world model. As the inaccuracy and computational cost of learned dynamics models increase with the prediction horizon, we are forced to optimize a fixed-horizon objective. This may reintroduce a slight degree of

myopia that could lead to actions which cause suboptimal behavior beyond the planning horizon. This, however, was not a practical issue in our empirical validation, and we expect our framework to further benefit as the accuracy of learned world models improves. From a theoretical standpoint, ZILOT induces a non-Markovian policy, even when expert trajectories are collected by a Markovian policy, and a Markovian policy would thus be sufficient for imitation. While the space of non-Markovian policies is larger, we find ZILOT to be able to efficiently find a near-optimal policy. This aligns with the fact that several practical zero-shot IL algorithms are based on efficient search over non-Markovian policies (e.g. those based on a goal-reaching policy and a classifier (Pathak et al., 2019; Pirotta et al., 2024)).

## Impact Statement

Advancements in imitation learning may lead to more capable robotic systems across a variety of application domains. While such systems could have societal implications depending on their use cases, our contributions are algorithmic rather than domain-specific.

## Acknowledgments

We thank Anselm Paulus, Mikel Zhobro, and Núria Armengol Urpí for their help throughout the project. Marco Bagatella and Jonas Frey are supported by the Max Planck ETH Center for Learning Systems. Georg Martius is a member of the Machine Learning Cluster of Excellence, EXC number 2064/1 – Project number 390727645. We acknowledge the support from the German Federal Ministry of Education and Research (BMBF) through the Tübingen AI Center (FKZ: 01IS18039A). Finally, this work was supported by the ERC – 101045454 REAL-RL.

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

# A. Additional Results

## A.1. Main Result Details

In tables 1 and 2 we provide detailed results for the figures 5 and 6. We also provide a summary of all planners we evaluated in figure 7.

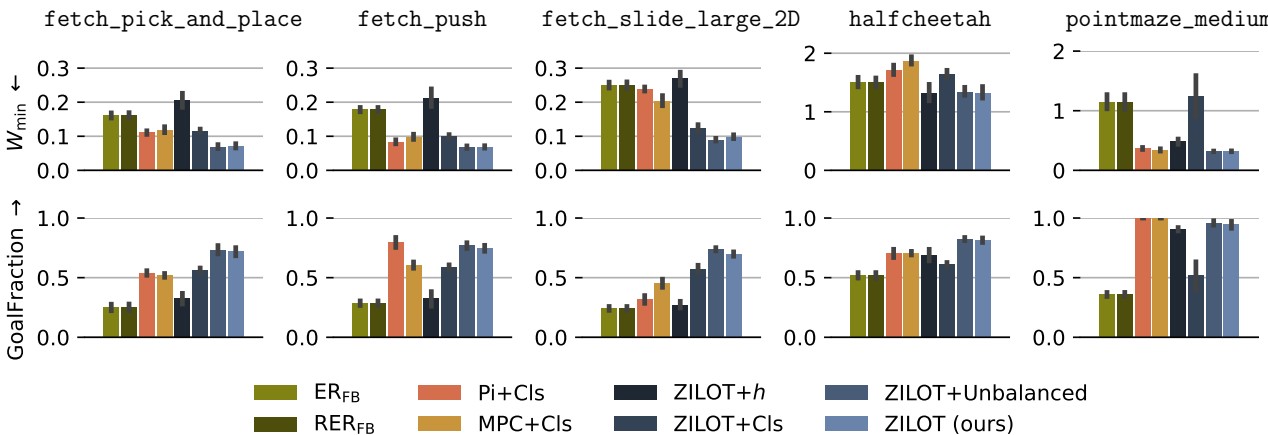

*Figure 7.* Summarized performance of all discussed Planners. See table 1 and table 2 for detailed results.

*Table 1.* Performance of Pi+Cls, MPC+Cls and ZILOT (ours) in all environments and tasks. Each metric is the mean over 20 trials, we then report the mean and standard deviation of those metrics across 5 seeds. We perform a Welch $t$-test with $p = 0.05$ do distinguish the best values and mark them bold. Values are rounded to 3 and 2 digits respectively.

| Task | $W_{min} \downarrow$ | | | | | GoalFraction $\uparrow$ | | | | |
|---|---|---|---|---|---|---|---|---|---|---|
| | ER$_{FB}$ | RER$_{FB}$ | Pi+Cls | MPC+Cls | ZILOT (ours) | ER$_{FB}$ | RER$_{FB}$ | Pi+Cls | MPC+Cls | ZILOT (ours) |
| fetch_pick_and_place-L-dense | 0.233±0.017 | 0.233±0.017 | 0.089±0.027 | 0.109±0.024 | **0.049±0.019** | 0.17±0.03 | 0.17±0.03 | 0.65±0.11 | 0.58±0.07 | **0.88±0.07** |
| fetch_pick_and_place-L-sparse | 0.170±0.011 | 0.170±0.011 | 0.112±0.014 | 0.127±0.022 | **0.092±0.015** | 0.35±0.04 | 0.35±0.04 | **0.62±0.05** | 0.43±0.04 | **0.65±0.05** |
| fetch_pick_and_place-S-dense | 0.183±0.019 | 0.183±0.019 | 0.113±0.022 | 0.101±0.022 | **0.049±0.014** | 0.16±0.04 | 0.16±0.04 | 0.41±0.07 | 0.62±0.08 | **0.85±0.08** |
| fetch_pick_and_place-S-sparse | 0.098±0.008 | 0.098±0.008 | **0.081±0.017** | 0.091±0.007 | 0.067±0.006 | 0.33±0.08 | 0.33±0.08 | 0.57±0.04 | 0.50±0.04 | **0.70±0.06** |
| fetch_pick_and_place-U-dense | 0.124±0.021 | 0.124±0.021 | 0.127±0.007 | 0.116±0.015 | **0.068±0.005** | 0.21±0.08 | 0.21±0.08 | 0.47±0.10 | 0.60±0.03 | **0.70±0.02** |
| fetch_pick_and_place-U-sparse | 0.163±0.028 | 0.163±0.028 | 0.142±0.005 | 0.160±0.008 | **0.098±0.003** | 0.30±0.07 | 0.30±0.07 | **0.51±0.02** | 0.38±0.03 | **0.55±0.05** |
| fetch_pick_and_place-all | 0.162±0.010 | 0.162±0.010 | 0.111±0.007 | 0.117±0.012 | **0.070±0.009** | 0.25±0.04 | 0.25±0.04 | 0.54±0.02 | 0.52±0.02 | **0.72±0.04** |
| fetch_push-L-dense | 0.246±0.001 | 0.246±0.001 | 0.056±0.001 | 0.085±0.018 | **0.041±0.015** | 0.15±0.00 | 0.15±0.00 | **0.96±0.03** | 0.72±0.09 | **0.91±0.06** |
| fetch_push-L-sparse | 0.184±0.014 | 0.184±0.014 | 0.101±0.011 | 0.103±0.010 | **0.082±0.004** | 0.35±0.04 | 0.35±0.04 | **0.65±0.06** | 0.44±0.04 | **0.69±0.06** |
| fetch_push-S-dense | 0.182±0.019 | 0.182±0.019 | 0.077±0.024 | 0.104±0.026 | **0.049±0.010** | 0.25±0.05 | 0.25±0.05 | **0.83±0.09** | 0.70±0.08 | **0.87±0.08** |
| fetch_push-S-sparse | 0.123±0.010 | 0.123±0.010 | **0.062±0.004** | 0.077±0.004 | **0.064±0.006** | 0.36±0.08 | 0.36±0.08 | **0.90±0.07** | 0.65±0.04 | 0.72±0.06 |
| fetch_push-U-dense | 0.141±0.011 | 0.141±0.011 | **0.102±0.044** | 0.091±0.009 | **0.065±0.004** | 0.28±0.07 | 0.28±0.07 | **0.72±0.18** | 0.67±0.08 | **0.77±0.02** |
| fetch_push-U-sparse | 0.195±0.024 | 0.195±0.024 | **0.106±0.014** | 0.131±0.012 | **0.109±0.007** | 0.32±0.03 | 0.32±0.03 | **0.70±0.12** | 0.45±0.05 | 0.53±0.03 |
| fetch_push-all | 0.178±0.008 | 0.178±0.008 | 0.084±0.007 | 0.098±0.010 | **0.068±0.005** | 0.29±0.02 | 0.29±0.02 | **0.79±0.05** | 0.61±0.03 | **0.75±0.03** |
| fetch_slide_large_2D-L-dense | 0.282±0.014 | 0.282±0.014 | 0.258±0.022 | 0.217±0.034 | **0.074±0.011** | 0.17±0.04 | 0.17±0.04 | 0.26±0.06 | 0.40±0.11 | **0.76±0.03** |
| fetch_slide_large_2D-L-sparse | 0.255±0.007 | 0.255±0.007 | 0.223±0.014 | 0.185±0.027 | **0.120±0.011** | 0.38±0.05 | 0.38±0.05 | 0.47±0.10 | **0.70±0.05** | **0.73±0.04** |
| fetch_slide_large_2D-S-dense | 0.232±0.029 | 0.232±0.029 | 0.299±0.006 | 0.254±0.022 | **0.111±0.010** | 0.19±0.05 | 0.19±0.05 | 0.21±0.10 | 0.31±0.06 | **0.51±0.07** |
| fetch_slide_large_2D-S-sparse | 0.215±0.014 | 0.215±0.014 | 0.266±0.006 | 0.230±0.021 | **0.086±0.015** | 0.28±0.04 | 0.28±0.04 | 0.31±0.02 | 0.43±0.02 | **0.74±0.04** |
| fetch_slide_large_2D-U-dense | 0.225±0.051 | 0.226±0.051 | 0.214±0.029 | 0.191±0.045 | **0.076±0.009** | 0.14±0.03 | 0.14±0.03 | 0.30±0.07 | 0.35±0.10 | **0.76±0.04** |
| fetch_slide_large_2D-U-sparse | 0.291±0.049 | 0.294±0.048 | 0.169±0.043 | 0.150±0.012 | **0.120±0.005** | 0.30±0.04 | 0.29±0.04 | 0.36±0.09 | 0.53±0.04 | **0.70±0.06** |
| fetch_slide_large_2D-all | 0.250±0.012 | 0.251±0.012 | 0.238±0.008 | 0.205±0.020 | **0.098±0.007** | 0.24±0.02 | 0.24±0.02 | 0.32±0.04 | 0.45±0.04 | **0.70±0.02** |
| halfcheetah-backflip | **2.002±0.149** | **2.002±0.149** | 3.089±0.588 | 4.281±0.371 | **2.625±0.780** | **0.44±0.08** | **0.44±0.08** | 0.28±0.13 | 0.12±0.12 | **0.57±0.17** |
| halfcheetah-backflip-running | 2.853±0.104 | 2.853±0.104 | 2.879±0.427 | 3.044±0.752 | **2.171±0.454** | 0.06±0.05 | 0.06±0.05 | 0.44±0.10 | **0.46±0.18** | **0.58±0.11** |
| halfcheetah-frontflip | **1.286±0.059** | **1.286±0.059** | 1.544±0.127 | 1.695±0.147 | **1.295±0.094** | 0.73±0.22 | 0.73±0.22 | 0.77±0.09 | 0.79±0.12 | **1.00±0.00** |
| halfcheetah-frontflip-running | **2.137±0.204** | **2.137±0.204** | 2.086±0.133 | 2.083±0.104 | **1.955±0.057** | 0.27±0.08 | 0.27±0.08 | 0.70±0.08 | **0.81±0.07** | **0.85±0.03** |
| halfcheetah-hop-backward | 0.910±0.316 | 0.910±0.316 | 0.806±0.110 | 0.950±0.075 | **0.589±0.107** | 0.65±0.19 | 0.65±0.19 | **0.96±0.03** | 0.90±0.02 | **0.96±0.03** |
| halfcheetah-hop-forward | **1.418±0.332** | **1.418±0.332** | 1.580±0.069 | 1.392±0.206 | **1.101±0.152** | 0.43±0.09 | 0.43±0.09 | **0.51±0.07** | 0.62±0.14 | **0.58±0.12** |
| halfcheetah-run-backward | 0.667±0.079 | 0.667±0.079 | 0.897±0.042 | 0.679±0.035 | **0.489±0.167** | 0.81±0.09 | 0.81±0.09 | **0.96±0.04** | **1.00±0.00** | **0.99±0.01** |
| halfcheetah-run-forward | 0.712±0.064 | 0.712±0.064 | 0.857±0.044 | 0.822±0.206 | **0.376±0.019** | 0.76±0.07 | 0.76±0.07 | **1.00±0.01** | 0.94±0.08 | **1.00±0.00** |
| halfcheetah-all | 1.498±0.105 | 1.498±0.105 | 1.717±0.101 | 1.868±0.079 | **1.325±0.123** | 0.52±0.03 | 0.52±0.03 | 0.70±0.05 | 0.71±0.02 | **0.82±0.02** |
| pointmaze_medium-circle-dense | 1.128±0.250 | 1.128±0.250 | 0.243±0.038 | 0.221±0.021 | **0.156±0.010** | 0.18±0.06 | 0.18±0.06 | **1.00±0.00** | **1.00±0.00** | **1.00±0.00** |
| pointmaze_medium-circle-sparse | 1.483±0.410 | 1.483±0.410 | **0.385±0.015** | **0.404±0.025** | 0.466±0.024 | 0.22±0.00 | 0.22±0.00 | **1.00±0.00** | **1.00±0.00** | 0.81±0.11 |
| pointmaze_medium-path-dense | 0.900±0.317 | 0.900±0.317 | 0.275±0.063 | 0.235±0.023 | **0.199±0.013** | 0.56±0.11 | 0.56±0.11 | **1.00±0.00** | **1.00±0.00** | **1.00±0.00** |
| pointmaze_medium-path-sparse | 1.086±0.505 | 1.086±0.505 | 0.555±0.080 | 0.511±0.035 | **0.459±0.015** | 0.48±0.10 | 0.48±0.10 | **1.00±0.00** | **1.00±0.00** | 0.97±0.03 |
| pointmaze_medium-all | 1.149±0.163 | 1.149±0.163 | 0.365±0.021 | 0.343±0.023 | **0.320±0.009** | 0.36±0.02 | 0.36±0.02 | **1.00±0.00** | **1.00±0.00** | 0.94±0.04 |

*Table 2.* Performance of our method and its ablations in all environments and tasks. Each metric is the mean over 20 trials, we then report the mean and standard deviation of those metrics across 5 seeds. We perform a Welch $t$-test with $p = 0.05$ do distinguish the best values and mark them bold. Values are rounded to 3 and 2 digits respectively.

| Task | $W_{min} \downarrow$ | | | | GoalFraction $\uparrow$ | | | |
|---|---|---|---|---|---|---|---|---|
| | ZILOT+$h$ | ZILOT+Cls | ZILOT+Unbalanced | ZILOT (ours) | ZILOT+$h$ | ZILOT+Cls | ZILOT+Unbalanced | ZILOT (ours) |
| fetch_pick_and_place-L-dense | 0.214±0.033 | 0.091±0.011 | **0.052±0.018** | **0.049±0.019** | 0.26±0.10 | 0.68±0.04 | **0.84±0.07** | **0.88±0.07** |
| fetch_pick_and_place-L-sparse | 0.188±0.014 | 0.158±0.004 | **0.095±0.016** | **0.092±0.015** | 0.40±0.01 | 0.35±0.02 | **0.65±0.08** | **0.65±0.05** |
| fetch_pick_and_place-S-dense | 0.198±0.042 | 0.089±0.019 | **0.045±0.006** | **0.049±0.014** | 0.36±0.15 | 0.71±0.07 | **0.86±0.03** | **0.85±0.08** |
| fetch_pick_and_place-S-sparse | 0.174±0.029 | 0.115±0.009 | **0.056±0.008** | 0.067±0.006 | 0.42±0.08 | 0.57±0.02 | **0.76±0.08** | 0.70±0.06 |
| fetch_pick_and_place-U-dense | 0.237±0.043 | 0.071±0.006 | **0.060±0.008** | 0.068±0.005 | 0.17±0.10 | **0.74±0.04** | **0.75±0.04** | 0.70±0.02 |
| fetch_pick_and_place-U-sparse | 0.229±0.034 | 0.167±0.004 | **0.101±0.008** | **0.098±0.003** | 0.34±0.04 | 0.33±0.05 | **0.54±0.05** | **0.55±0.05** |
| fetch_pick_and_place-all | 0.207±0.026 | 0.115±0.007 | **0.068±0.008** | 0.070±0.009 | 0.32±0.06 | 0.56±0.02 | **0.73±0.05** | 0.72±0.04 |
| fetch_push-L-dense | 0.211±0.020 | 0.071±0.006 | **0.040±0.004** | 0.041±0.015 | 0.27±0.06 | 0.73±0.02 | **0.91±0.03** | **0.91±0.06** |
| fetch_push-L-sparse | 0.200±0.022 | 0.150±0.005 | 0.101±0.014 | **0.082±0.004** | 0.39±0.06 | 0.36±0.03 | **0.65±0.07** | **0.69±0.06** |
| fetch_push-S-dense | 0.203±0.046 | 0.077±0.008 | **0.049±0.010** | **0.049±0.010** | 0.32±0.14 | 0.72±0.05 | **0.86±0.05** | **0.87±0.08** |
| fetch_push-S-sparse | 0.197±0.055 | 0.097±0.006 | **0.060±0.009** | 0.064±0.006 | 0.40±0.17 | 0.56±0.02 | **0.78±0.06** | 0.72±0.06 |
| fetch_push-U-dense | 0.228±0.045 | 0.068±0.007 | **0.058±0.009** | 0.065±0.004 | 0.20±0.10 | **0.78±0.04** | **0.81±0.03** | 0.77±0.02 |
| fetch_push-U-sparse | 0.224±0.047 | 0.136±0.017 | **0.100±0.007** | 0.109±0.007 | 0.36±0.07 | 0.39±0.05 | **0.61±0.05** | 0.53±0.03 |
| fetch_push-all | 0.211±0.033 | 0.100±0.006 | **0.068±0.005** | **0.068±0.005** | 0.32±0.08 | 0.59±0.02 | **0.77±0.03** | 0.75±0.03 |
| fetch_slide_large_2D-L-dense | 0.255±0.022 | 0.098±0.027 | **0.060±0.009** | 0.074±0.011 | 0.26±0.08 | 0.69±0.08 | **0.81±0.07** | 0.76±0.03 |
| fetch_slide_large_2D-L-sparse | 0.236±0.020 | 0.181±0.039 | **0.112±0.016** | **0.120±0.011** | 0.41±0.04 | 0.45±0.08 | **0.83±0.08** | 0.73±0.04 |
| fetch_slide_large_2D-S-dense | 0.256±0.035 | 0.105±0.011 | **0.091±0.009** | 0.111±0.010 | 0.23±0.10 | **0.63±0.03** | 0.59±0.10 | 0.51±0.07 |
| fetch_slide_large_2D-S-sparse | 0.272±0.045 | 0.132±0.033 | **0.084±0.010** | 0.086±0.015 | 0.28±0.07 | 0.52±0.08 | **0.79±0.04** | 0.74±0.04 |
| fetch_slide_large_2D-U-dense | 0.315±0.051 | 0.087±0.009 | **0.074±0.011** | 0.076±0.004 | 0.12±0.08 | **0.75±0.07** | **0.75±0.04** | **0.76±0.04** |
| fetch_slide_large_2D-U-sparse | 0.288±0.058 | 0.147±0.009 | **0.117±0.008** | 0.120±0.005 | 0.30±0.04 | 0.41±0.04 | **0.68±0.07** | **0.70±0.06** |
| fetch_slide_large_2D-all | 0.270±0.025 | 0.125±0.011 | **0.090±0.005** | 0.098±0.007 | 0.27±0.04 | 0.57±0.04 | **0.74±0.02** | 0.70±0.02 |
| halfcheetah-backflip | **1.947±0.312** | 3.170±0.730 | 2.710±0.742 | **2.625±0.780** | **0.50±0.18** | 0.43±0.14 | **0.55±0.20** | **0.57±0.17** |
| halfcheetah-backflip-running | **2.537±0.810** | **2.479±0.284** | **2.297±0.525** | **2.171±0.454** | **0.47±0.27** | **0.50±0.11** | **0.58±0.16** | **0.58±0.11** |
| halfcheetah-frontflip | **1.172±0.091** | 1.796±0.173 | 1.330±0.168 | 1.295±0.094 | 0.96±0.03 | 0.52±0.03 | **0.98±0.03** | **1.00±0.00** |
| halfcheetah-frontflip-running | 2.526±0.110 | **2.091±0.210** | 1.969±0.075 | 1.955±0.057 | 0.13±0.07 | 0.60±0.06 | **0.88±0.09** | 0.85±0.03 |
| halfcheetah-hop-backward | **0.739±0.736** | 0.889±0.103 | **0.548±0.056** | 0.589±0.107 | **0.84±0.33** | 0.82±0.07 | **0.96±0.04** | **0.96±0.03** |
| halfcheetah-hop-forward | **0.682±0.120** | 1.070±0.086 | 1.007±0.094 | 1.101±0.152 | **0.78±0.12** | 0.63±0.08 | **0.67±0.07** | 0.58±0.12 |
| halfcheetah-run-backward | **0.555±0.415** | 0.838±0.139 | **0.473±0.162** | **0.489±0.167** | **0.92±0.11** | 0.68±0.03 | **0.99±0.01** | **0.99±0.01** |
| halfcheetah-run-forward | **0.372±0.156** | 0.742±0.044 | **0.381±0.026** | **0.376±0.019** | **0.93±0.09** | 0.72±0.05 | **1.00±0.01** | **1.00±0.00** |
| halfcheetah-all | **1.316±0.181** | 1.634±0.089 | **1.339±0.090** | **1.325±0.123** | 0.69±0.06 | 0.61±0.02 | **0.83±0.02** | 0.82±0.02 |
| pointmaze_medium-circle-dense | 0.252±0.032 | 0.651±0.377 | **0.168±0.015** | **0.156±0.010** | 0.91±0.04 | 0.62±0.25 | **1.00±0.00** | **1.00±0.00** |
| pointmaze_medium-circle-sparse | **0.465±0.056** | 1.074±0.115 | **0.465±0.028** | **0.466±0.024** | **0.87±0.03** | 0.41±0.10 | **0.83±0.10** | **0.81±0.11** |
| pointmaze_medium-path-dense | 0.495±0.130 | 1.835±1.064 | **0.192±0.008** | 0.199±0.013 | 0.95±0.03 | 0.45±0.29 | **1.00±0.00** | **1.00±0.00** |
| pointmaze_medium-path-sparse | 0.716±0.119 | 1.416±0.828 | **0.444±0.010** | 0.459±0.015 | 0.89±0.10 | 0.61±0.24 | **0.99±0.01** | 0.97±0.03 |
| pointmaze_medium-all | 0.482±0.055 | 1.244±0.463 | **0.317±0.008** | 0.320±0.009 | 0.91±0.02 | 0.52±0.15 | **0.95±0.03** | 0.94±0.04 |

## A.2. Finite Horizon Ablations

As discussed in section 4, we are forced to optimize the objective over a finite horizon $H$ due to the imperfections in the learned dynamics model and computational constraints. The hyperparameter $H$ should thus be as large as possible, as long as the model remains accurate. We visualize this trade-off in figure 8 for environment fetch_slide_large_2D. It is clearly visible that if the horizon is smaller than 16, the value we chose for our experiments, then performance rapidly deteriorates towards the one of the myopic planners. However, when increasing the horizon beyond 16, performance does not improve, suggesting that the model is not accurate enough to plan beyond this horizon.

## A.3. Single Goal Performance

When the expert trajectory consists of only a single goal, myopic planning is of course sufficient to imitate the expert. To verify this we evaluate the performance of all planners in the standard single goal task of the environments. Figure 9 shows the success rate of all planners in this task verifying that non-myopic planning neither hinders nor helps in this case.

## A.4. Environment Similarity

We evaluate ZILOT and the myopic baselines Pi+Cls and MPC+Cls on the walker environment (Tassa et al., 2018) in Table 3. Because this environment is very similar to the halfcheetah environment we use in our main evaluation, we can reuse the same goal-space, tasks, and data collection method. These similarities are also visible in the performance of the three methods.

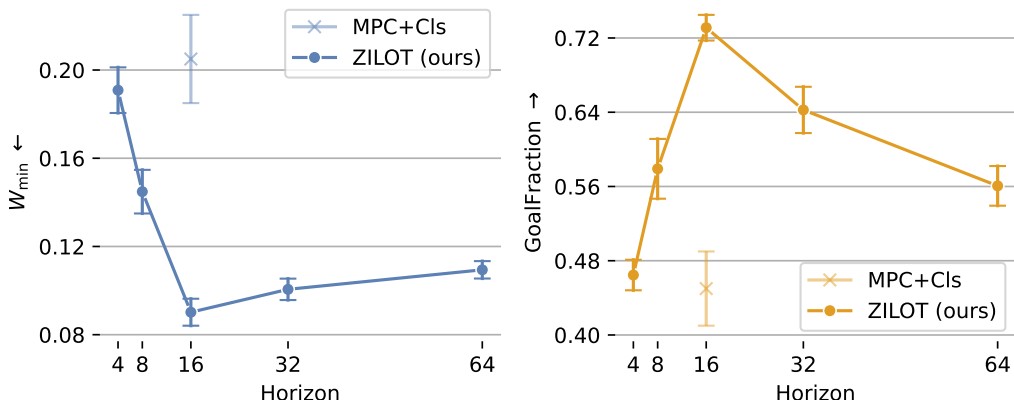

*Figure 8.* Mean performance across five seeds in `fetch_slide_large_2D` for different planning horizons.

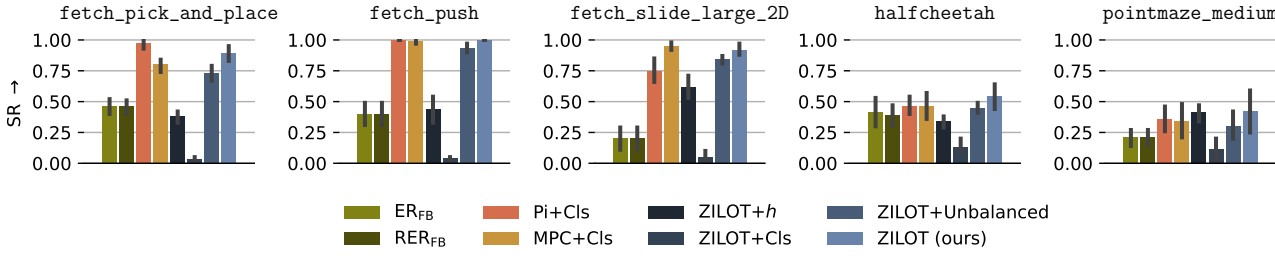

*Figure 9.* Single Goal Success Rate in the standard single goal tasks of the environments. We report the mean performance across 20 trials and standard deviationacross 5 seeds.

*Table 3.* Evaluation on the `walker` environment with the `halfcheetah` results repeated for comparison.

| Task | $W_{\min}\downarrow$ | | | GoalFraction $\uparrow$ | | |
|---|---|---|---|---|---|---|
| | Pi+Cls | MPC+Cls | ZILOT (ours) | Pi+Cls | MPC+Cls | ZILOT (ours) |
| `walker-backflip` | 2.804±0.056 | 1.737±0.146 | **1.273±0.205** | 0.34±0.07 | **0.89±0.03** | **0.92±0.06** |
| `walker-backflip-running` | 3.039±0.292 | 2.444±0.189 | **1.709±0.093** | 0.49±0.07 | 0.70±0.08 | **0.81±0.09** |
| `walker-frontflip` | 2.688±0.400 | 1.830±0.185 | **1.551±0.086** | 0.57±0.16 | **0.94±0.04** | **0.95±0.07** |
| `walker-frontflip-running` | 2.597±0.265 | **1.937±0.172** | **1.921±0.149** | 0.55±0.03 | 0.63±0.16 | **0.76±0.11** |
| `walker-hop-backward` | 1.447±0.076 | **0.872±0.032** | **0.836±0.100** | 0.64±0.11 | **0.78±0.06** | **0.84±0.07** |
| `walker-hop-forward` | 0.932±0.098 | 0.663±0.071 | **0.467±0.044** | 0.95±0.05 | **0.99±0.01** | **1.00±0.01** |
| `walker-run-backward` | 1.290±0.148 | **1.050±0.086** | **0.957±0.111** | 0.81±0.08 | 0.83±0.14 | **0.84±0.09** |
| `walker-run-forward` | 1.180±0.105 | 0.954±0.079 | **0.672±0.058** | 0.86±0.07 | **0.97±0.03** | **0.99±0.01** |
| `walker-all` | 1.997±0.047 | 1.436±0.049 | **1.174±0.061** | 0.65±0.03 | 0.84±0.02 | **0.89±0.04** |
| `halfcheetah-backflip` | **3.089±0.588** | 4.281±0.371 | **2.625±0.780** | 0.28±0.13 | 0.12±0.12 | **0.57±0.17** |
| `halfcheetah-backflip-running` | 2.879±0.427 | 3.044±0.752 | **2.171±0.454** | 0.44±0.10 | **0.46±0.18** | **0.58±0.11** |
| `halfcheetah-frontflip` | 1.544±0.127 | 1.695±0.147 | **1.295±0.094** | 0.77±0.09 | 0.79±0.12 | **1.00±0.00** |
| `halfcheetah-frontflip-running` | 2.086±0.133 | 2.083±0.104 | **1.955±0.057** | 0.70±0.08 | **0.81±0.07** | **0.85±0.03** |
| `halfcheetah-hop-backward` | 0.806±0.110 | 0.950±0.075 | **0.589±0.107** | **0.96±0.03** | 0.90±0.02 | **0.96±0.03** |
| `halfcheetah-hop-forward` | 1.580±0.069 | 1.392±0.206 | **1.101±0.152** | 0.51±0.07 | **0.62±0.14** | 0.58±0.12 |
| `halfcheetah-run-backward` | 0.897±0.092 | 0.679±0.035 | **0.489±0.167** | 0.96±0.04 | **1.00±0.00** | **0.99±0.01** |
| `halfcheetah-run-forward` | 0.857±0.044 | 0.822±0.206 | **0.376±0.019** | **1.00±0.01** | 0.94±0.08 | **1.00±0.00** |
| `halfcheetah-all` | 1.717±0.101 | 1.868±0.079 | **1.325±0.123** | 0.70±0.05 | 0.71±0.02 | **0.82±0.02** |

## B. Forward-Backward Representations and Imitation Learning

In a foundational paper in zero-shot, model-free IL, (Pirotta et al., 2024) propose several methods based on the forward-backward (FB) framework  (Touati & Ollivier, 2021).  FB trains two functions $F$ and $B$, which recover a low-rank approximation of the successor measure, as well as a parameterized policy $(\pi_z)_{z\in\mathbb{R}^d}$. These functions can be trained offline, without supervision, so that for each reward $r$, an optimal policy $\pi_{z_r}$ can be recovered. This property gives rise to a range of reward-based and occupancy-matching based methods for zero-shot IL. In the following we will go over each method, and

discuss how it differs from ZILOT in terms of objective. We will highlight how several methods do not directly apply to our setting, which involves expert demonstrations that are actionless, rough, and partial. We refer the reader to section C.10 for implementation details of baselines based on FB.

## B.1. FB Imitation Learning Approaches

**Behavioral Cloning**   The first approach in Pirotta et al. (2024) is is based on gradient descent on the latent $z$ to find the policy $\pi_z$ that maximizes the likelihood of expert actions. Since this approach strictly requires expert actions it does not apply in our case.

**Reward-Based Imitation Learning**   (Pirotta et al., 2024) derive two reward-based zero-shot IL methods maximizing the reward $r(\cdot) = \rho^E(\cdot)/\rho^{\mathcal{D}_\beta}(\cdot)$ (ER$_{\text{FB}}$) (Ma et al., 2022; Kim et al., 2022) and its regularized counterpart $r(\cdot) = \rho^E(\cdot)/(\rho^E(\cdot) + \rho^{\mathcal{D}_\beta}(\cdot))$ (RER$_{\text{FB}}$) (Reddy et al., 2020; Zolna et al., 2020). While ZILOT's objective is based on a Wasserstein distance, these rewards are derived from *regularized $f$-divergence* objectives. These objectives are fortunately tractable, and can be minimized by solving an RL problem with additive rewards. In practice, this corresponds to assigning a scalar reward to each state visited by the expert, without considering the order of the states in the expert trajectory. However, as stated in Section 4.2 of Pirotta et al. (2024), this regularization comes at a cost, particularly if the state does not contain dynamical information, or in ergodic MDPs. In this case, a policy can maximize the reward by remaining in the most likely expert state, and the objective might be optimized by degenerate solution. On the other hand, such solution would be discarded by ZILOT, which uses an unregularized objective.

Nonetheless, these two instantiations are fully compatible with partial and rough demonstrations. Thus, we provide an empirical comparison in Section 5.

**Distribution Matching**   A further approach in Pirotta et al. (2024) finds the policy $\pi_z$ whose occupancy matches the expert occupancy w.r.t. different distances on the space of measures. ZILOT also performs occupancy matching, but with respect to Wasserstein distances. However, ZILOT is designed to handle state abstraction, i.e. partial states. To the best of our understanding, distribution- and feature-matching flavors of FB-IL require the demonstration to contain full states, unless further FB representations are trained to approximate successor measures over abstract states. While the standard implementation of distribution-matching FB-IL cannot imitate rough demonstrations, we believe that an extension in this direction may be interesting for future work.

**Goal-Based Imitation**   Pirotta et al. (2024) also instantiate a hierarchical, goal-based imitation method, in which the FB framework is only used for goal-reaching. This idea is closely related with one of our baselines (Pi+Cls). However, their framework assumes that trajectories to imitate are not rough and, instead of using a classifier, the goal can be chosen at a fixed offset of in time for each time-step. In any case, their approach remains myopic as per Proposition 3.1. Empirically, Pirotta et al. (2024) observe that this instantiation of FB-IL does not significantly outperform an equivalent method relying on TD3+HER instead. As the latter method is very similar to our Pi+Cls baseline, we do not investigate this approach further in this work.

# C. Implementation Details

## C.1. ZILOT

The proposed method is motivated and explained in section 4. We now present additional details.

**Sinkhorn**   First, we rescale the matrix $\boldsymbol{C}$ by $T_{\max}$ and clamp it to the range $[0, 1]$ before running Sinkhorns algorithm. The precise operation performed is

$$\boldsymbol{C} \leftarrow \min\left(1, \max(0, \boldsymbol{C}/T_{\max})\right). \tag{19}$$

This is done so that the same entropy regularization $\epsilon$ can be used across all environments, and to ensure there are no outliers that hinder the convergence of the Sinkhorn algorithm. For the algorithm itself, we use a custom implementation for batched OT computation, heavily inspired by (Flamary et al., 2021) and (Cuturi et al., 2022). We run our Sinkhorn algorithm for $r = 500$ iterations with a regularization factor of $\epsilon = 0.02$.

**Truncation**   When the agent gets close to the end of the expert trajectory, then we might have that $t_K < k + H$, i.e. the horizon is larger than needed. We thus truncate the planning horizon to the estimated remaining number of steps (and at least 1), i.e. we set

$$H_{\text{actual}} \leftarrow \max\left(1, \min(t_K - k, H)\right). \tag{20}$$

**Unbalanced OT**   As mentioned in the main text in section 5.3, we can use unbalanced OT (Liero et al., 2018; Séjourné et al., 2019) to address that fact that the uniform marginal for the goal occupancy approximation may not be feasible. Unbalanced OT replaces this hard constraint of $\boldsymbol{T}^\top \cdot \boldsymbol{1}_N = \boldsymbol{1}_M$ into the term $\xi_b \text{KL}(\boldsymbol{T}^\top \cdot \boldsymbol{1}_N, \boldsymbol{1}_M)$ in the objective function. For our experiments we have chosen $\xi_b = 1$.

## C.2. TD-MPC2 Modifications

As TD-MPC2 (Hansen et al., 2024) is already a multi-task algorithm that is conditioned on a learned task embedding $t$ from a task id $i$, we only have to switch out this conditioning to a goal latent $z_g$ to arrive at a goal-conditioned algorithm as detailed in table 4. We remove the conditioning on the encoders and the dynamics model $f$ completely as the goal conditioning of GC-RL only changes the reward but not the underlying Markov Decision Process $\mathcal{M}$ (assuming truncation after goal reaching, see section 2.3). For training we adopt all TD-MPC2 hyperparameters directly (see table 7). As mentioned in the main text, we also train a small MLP to predict $W$ that regresses on $V$.

Table 4. Our modifications to TD-MPC2 to making it goal- instead of task-conditioned.

|  | TD-MPC2 (Hansen et al., 2024) | "GC"-TD-MPC2 (our changes) |
| --- | --- | --- |
| Task/Goal Embedding | $t = E(i)$ | $z_g = h_g(g)$ |
| Encoder | $z = h(s, t)$ | $z = h(s)$ |
| Dynamics | $z' = f(z, a, t)$ | $z' = f(z, a)$ |
| Reward Prediction | $r = R(z, a, t)$ | $r = R(z, a, z_g)$ |
| $Q$-function | $q = Q(z, a, t)$ | $q = Q(z, a, z_g)$ |
| Policy | $a \sim \pi(z, t)$ | $a \sim \pi(z, z_g)$ |

We have found the computation of pair-wise distances $d$ to be the major computational bottleneck in our method, as TD-MPC2 computes them as $d = -V^\pi(s, g) = -Q(z, \pi(z, z_g), z_g)$ where $z = h(s), z_g = h_g(g)$. To speed-up computation, we train a separate network that estimates the value function directly. It employs a two-stream architecture (Schaul et al., 2015; Eysenbach et al., 2022) of the form $V^\pi(z, z_g) = \phi(z)^\top \psi(z_g)$ where $\phi$ and $\psi$ are small MLPs for fast inference of pair-wise distances.

Our GC-TD-MPC2 is trained like the original TD-MPC2 with two losses additionally employing HER (Andrychowicz et al., 2017) to sample goals $g$ which we discuss in detail in Appendix C.4. The first loss combines a multi-step loss for $d$ and $h$ with a single-step TD-step loss for $R$ and $Q$

$$\mathcal{L} = \mathbb{E}_{\substack{(s,a,s')_{0:H} \sim \mathcal{D} \\ g_{0:H} \sim \text{HER}_\gamma(s_{0:H})}} \left[ \sum_{t=0}^{H} \lambda^t \left( \|z_t' - \text{sg}(h(s_t'))\|_2^2 + \text{CE}(R(z_t, a_t, z_{g_t}), r_t) + \text{CE}(Q(z_t, a_t, z_{g_t}), q_t) \right) \right] \tag{21}$$

where sg is the "stop-gradient"-operator, $z_t, z_{g_t}$, and $z_t'$ are defined in Table 4, rewards are $r_t = \mathbb{I}_{s_t = g_t} - 1$, and (undiscounted) TD-targets are $q_t = \max(r_t + \mathbb{I}_{s_t \neq g_t} \cdot \overline{Q}(z_t', \pi(z_t', z_{g_t}), z_{g_t}), -T_{\max})$. The second loss is a SAC-style loss for $\pi$ (Haarnoja et al., 2018)

$$\mathcal{L}_\pi = \mathbb{E}_{\substack{(s,a,s')_{0:H} \sim \mathcal{D} \\ g_{0:H} \sim \text{HER}_\gamma(s_{0:H})}} \left[ \sum_{t=0}^{H} \lambda^t \left( \alpha Q(z_t, \pi(z_t, z_{g_t}), z_{g_t}) + \beta \mathcal{H}(\pi(\cdot | z_t, z_{g_t})) \right) \right], z_t = d(z_t, a_t), z_0 = h(s_0). \tag{22}$$

Additional to GC-TD-MPC2 we then also train our goal-conditioned value function $V^\pi(z, z_g) = \phi(z)^\top \psi(z_g)$ using the same TD-targets as in Equation (21)

$$\mathcal{L}_{\phi,\psi} = \mathbb{E}_{\substack{s_{0:H} \sim \mathcal{D} \\ g_{0:H} \sim \text{HER}_\gamma(s_{0:H})}} \left[ \sum_{t=0}^{H} \lambda^t \left( \phi(z_t)^\top \psi(z_{g_t}) - q_t \right)^2 \right]. \tag{23}$$

## C.3. Runtime

ZILOT runs at 2 to 4Hz on an Nvidia RTX 4090 GPU, depending on the size of $H$ and the size of the OT problem. Given that the MPC+Cls method runs at around 25 to 72Hz with the same networks and on the same hardware, it is clear that most computation is spent on preparing the cost-matrix $C$ and running the Sinkhorn solver. Several further steps could be taken to speed-up the Sinkhorn algorithm itself, including $\eta$-schedules and/or Anderson acceleration (Cuturi et al., 2022) as well as warm-starting it with potentials, e.g. from previous (optimizer) steps or from a trained network (Amos et al., 2023).

## C.4. Goal Sampling

As mentioned in the main text, we follow prior work (Andrychowicz et al., 2017; Bagatella & Martius, 2023; Tian et al., 2021) and sample goals from the future part of trajectories in $\mathcal{D}_\beta$ in order to synthesize rewards without supervision. The exact procedure is as follows:

- With probability $p_{\text{future}} = 0.6$ we sample a goal from the future part of the trajectory with time offset $t_\Delta \sim \text{Geom}(1-\gamma)$.
- With probability $p_{\text{next}} = 0.2$ we sample the next goal in the trajectory.
- With probability $p_{\text{rand}} = 0.2$ we sample a random goal from the dataset.

## C.5. Training

We train our version of TD-MPC2 offline with the datasets detailed in table 5 for 600k steps. Training took about 8 to 9 hours on a single Nvidia A100 GPU. Note that as TD-MPC2 samples batches of 3 transitions per element, we effectively sample $3 \cdot 256 = 768$ transitions per batch. The resulting models are then used for all planners and experiments.

*Table 5.* Environment description. We detail the datasets used for training.

| Environment | Dataset | #Transitions |
|---|---|---|
| fetch_push | WGCSL (Yang et al., 2022) (expert+random) | 400k + 400k |
| fetch_pick_and_place | WGCSL (Yang et al., 2022) (expert+random) | 400k + 400k |
| fetch_slide_large_2D | custom (curious exploration (Pathak et al., 2019)) | 500k |
| halfcheetah | custom (curious exploration (Pathak et al., 2019)) | 500k |
| pointmaze_medium | D4RL (Fu et al., 2021) (expert) | 1M |

## C.6. Environments

We provide environment details in table 6. Note that while we consider an undiscounted setting, we specify $\gamma$ for the goal sampling procedure above.

*Table 6.* Environment details. We detail the goal abstraction $\phi$, metric $h$, threshold $\epsilon$, horizon $H$, maximum episode length $T_{\max}$, and discount factor $\gamma$ used for each environment.

| Environment | Goal Abstraction $\phi$ | Metric $h$ | Threshold $\epsilon$ | Horizon $H$ | $T_{\max}$ | $\gamma$ |
|---|---|---|---|---|---|---|
| fetch_push | $(x, y, z)_{\text{cube}}$ | $\|\cdot\|_2$ | 0.05 | 16 | 50 | 0.975 |
| fetch_pick_and_place | $(x, y, z)_{\text{cube}}$ | $\|\cdot\|_2$ | 0.05 | 16 | 50 | 0.975 |
| fetch_slide_large_2D | $(x, y, z)_{\text{cube}}$ | $\|\cdot\|_2$ | 0.05 | 16 | 50 | 0.975 |
| halfcheetah | $(x, \theta_y)$ | $\|\cdot\|_2$ | 0.50 | 32 | 200 | 0.990 |
| pointmaze_medium | $(x, y)$ | $\|\cdot\|_2$ | 0.45 | 64 | 600 | 0.995 |

The environments fetch_push and fetch_pick_and_place and pointmaze_medium are used as is. As halfcheetah is not goal-conditioned by default, we define our own goal range to be $(x, \theta_y) \in [-5, 5] \times [-4\pi, 4\pi]$[3]. fetch_slide_large_2D is a variation of the fetch_slide environment where the table size exceeds the arm's range and the arm is restricted to two-dimensional movement touching the table.

---

[3]Note that the halfcheetah environment does not reduce $\theta$ with any kind of modular operation, i.e. states with $\theta = 0$ and $\theta = 2\pi$ are distinct.

## C.7. Tasks

The tasks for the `fetch` and `pointmaze` environments are specified in the environments normal goal-space. Their shapes can be seen in the figures in appendix E. To make the tasks for `halfcheetah` more clear, we visualize some executions of our method in the figures 10, 11, 12, 13, 14, and 15.

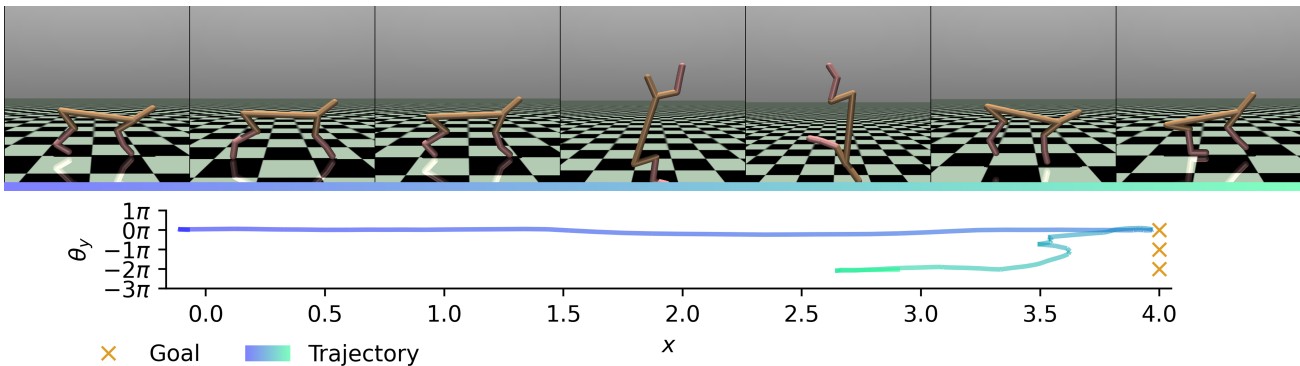

*Figure 10.* Example trajectory of ZILOT (ours) in `halfcheetah-backflip-running`.

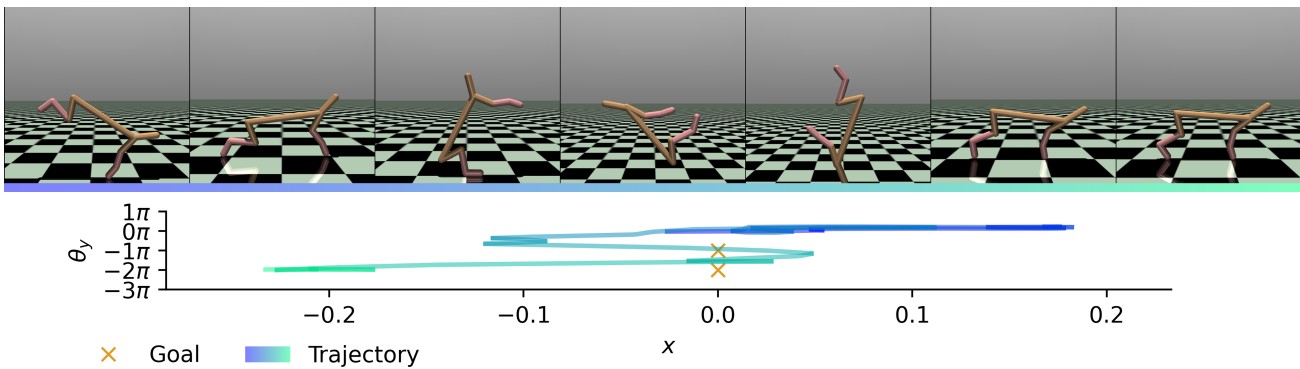

*Figure 11.* Example trajectory of ZILOT (ours) in `halfcheetah-backflip`.

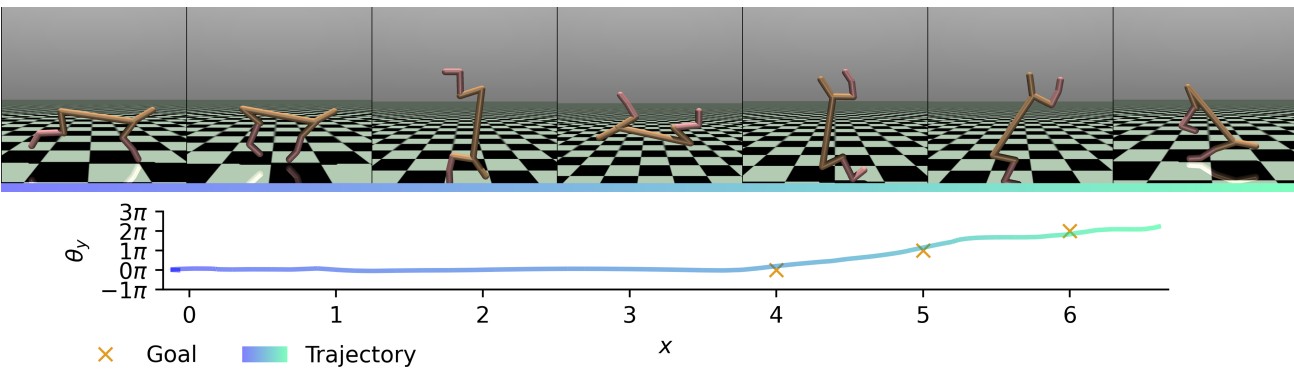

*Figure 12.* Example trajectory of ZILOT (ours) in `halfcheetah-frontflip-running`.

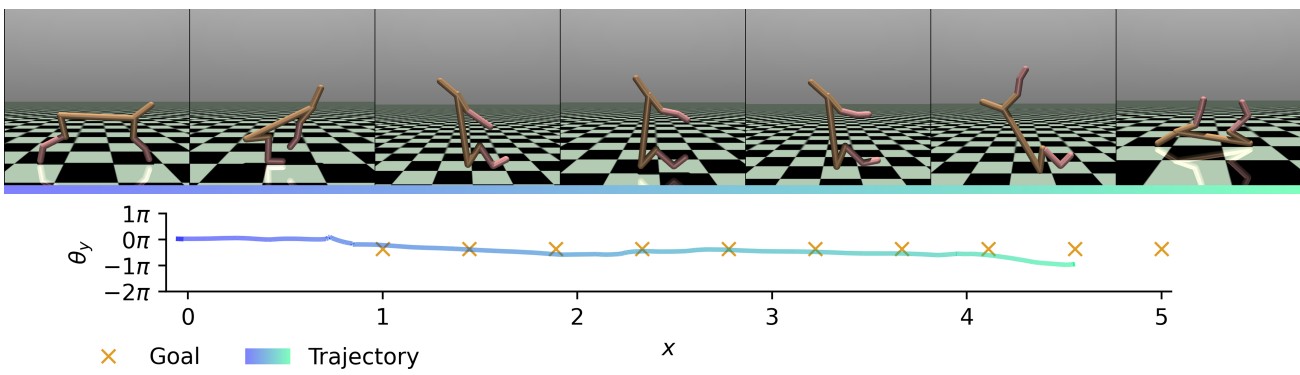

*Figure 15.* Example trajectory of ZILOT (ours) in `halfcheetah-hop-forward`.

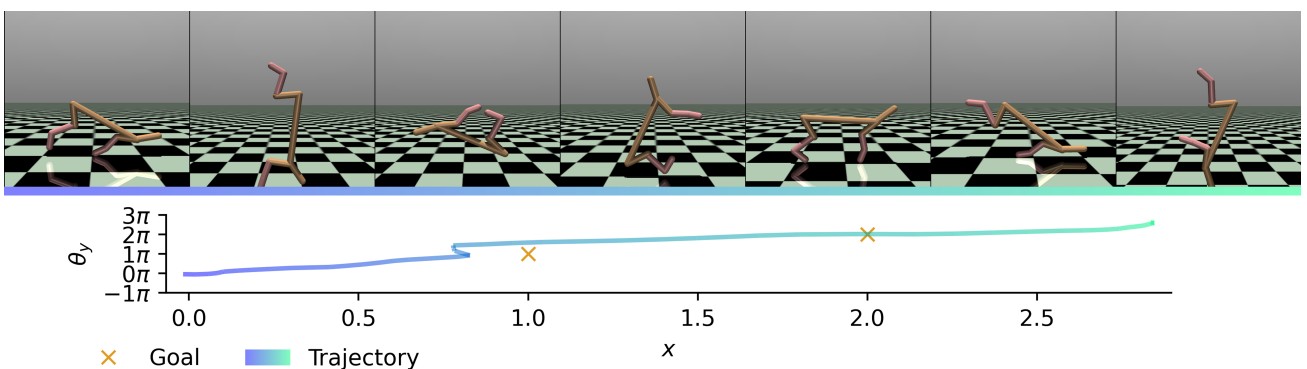

*Figure 13.* Example trajectory of ZILOT (ours) in `halfcheetah-frontflip`.

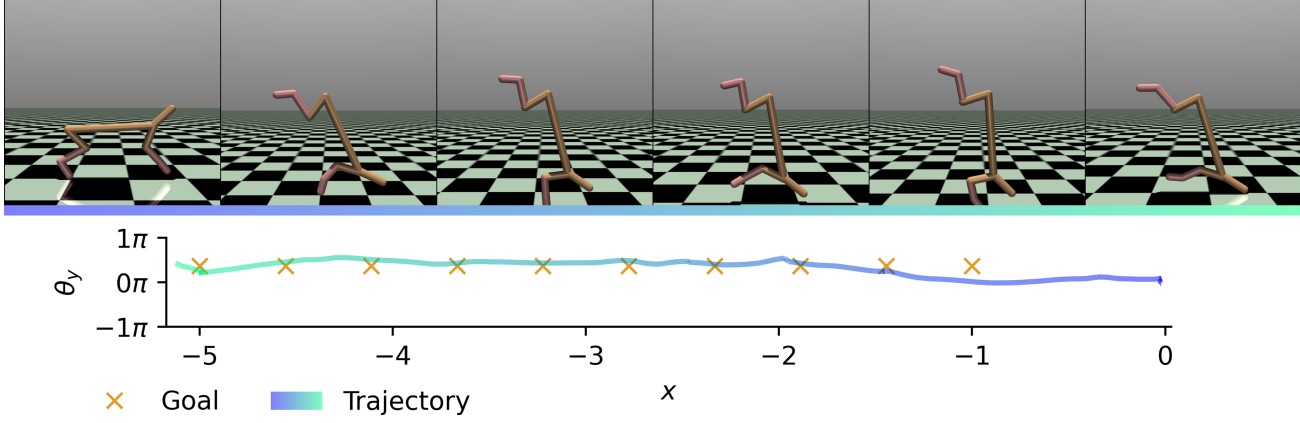

*Figure 14.* Example trajectory of ZILOT (ours) in `halfcheetah-hop-backward`.

## C.8. Task Difficulty

This section investigates the ability of ZILOT to imitate trajectories that do not appear in the offline dataset it is trained on. As ZILOT uses a learned dynamics model and an off-policy value function, it should in theory be able to stitch together any number of trajectories in the dataset. To get some qualitative intuition we overlay the following: first, a kernel density estimate of the data distribution in the offline datasets, second, an expert trajectory to imitate, and finally the five trajectories that are closest to the expert w.r.t. the Wasserstein distance under the goal-metric $h$. We present a few tasks for each environment in Figures 16, 18, 19, 17, and 20.

Comparing the density estimates and the expert trajectories, we can see that essentially all expert trajectories are within distribution. Although, especially in `halfcheetah`, there are some tasks, such as `hop-forward` and `backflip-running` with very little coverage which might explain the bad performance of all planners in these tasks (see table 1). Comparing the selected trajectories with the expert trajectory, it is also evident that the expert demonstrations are not directly present in the datasets. Thus, ZILOT is capable of imitating unseen *sequences* of states, as long as each individual state is within the support of the training data. In other words, ZILOT is capable of off-policy learning, or trajectory stitching.

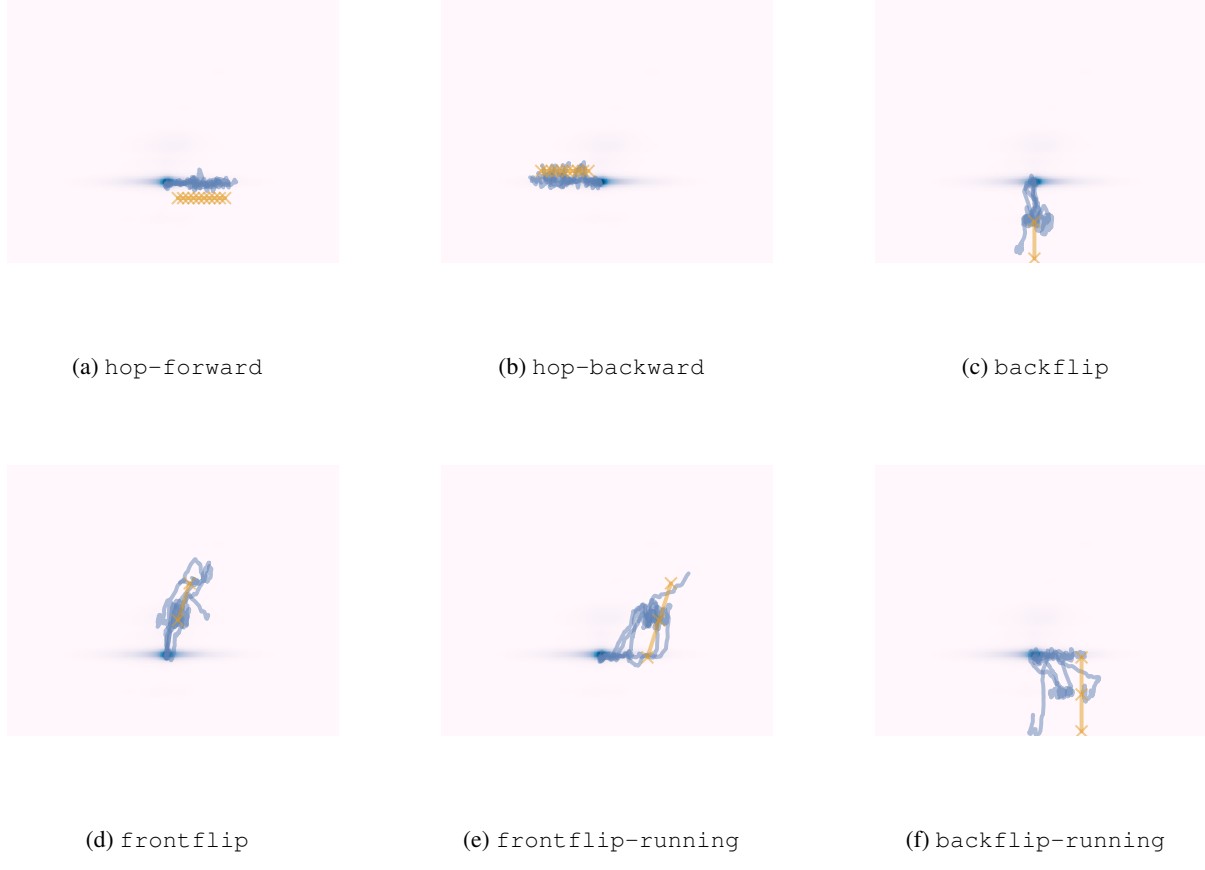

(a) `hop-forward`      (b) `hop-backward`      (c) `backflip`

(d) `frontflip`      (e) `frontflip-running`      (f) `backflip-running`

*Figure 16.* The 5 trajectories (blue) from the dataset that are closest to the expert trajectory in different `halfcheetah` tasks (orange) overlayed over a kernel density estimate of the goal occupancy in the full training dataset.

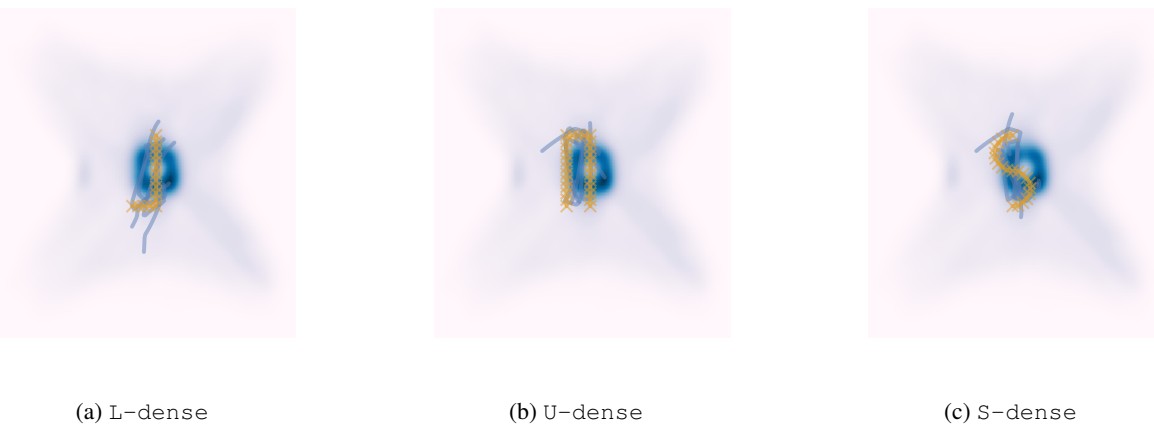

(a) `L-dense`         (b) `U-dense`         (c) `S-dense`

*Figure 17.* The 5 trajectories (blue) from the dataset that are closest to the expert trajectory in different `fetch_slide_large_2D` tasks (orange) overlayed over a kernel density estimate of the goal occupancy in the full training dataset.

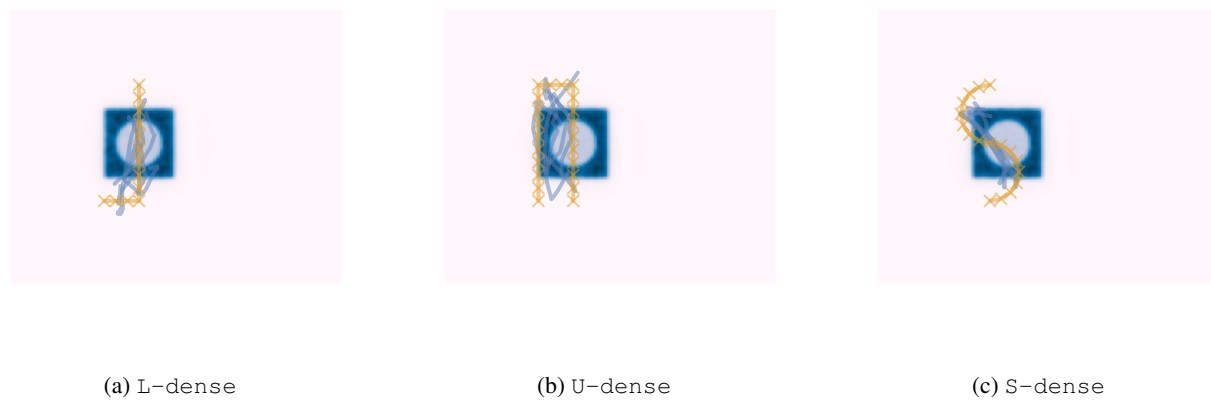

(a) `L-dense`         (b) `U-dense`         (c) `S-dense`

*Figure 18.* The 5 trajectories (blue) from the dataset that are closest to the expert trajectory in different `fetch_push` tasks (orange) overlayed over a kernel density estimate of the goal occupancy in the full training dataset.

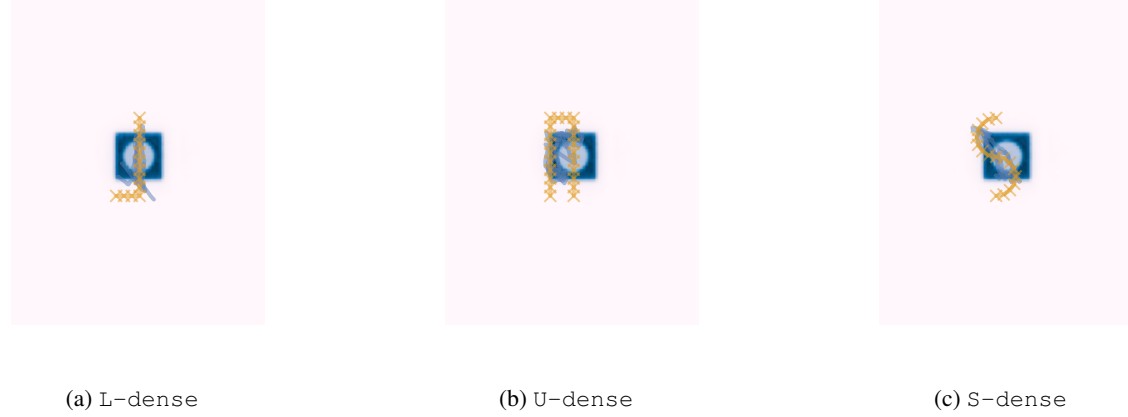

(a) `L-dense`         (b) `U-dense`         (c) `S-dense`

*Figure 19.* The 5 trajectories (blue) from the dataset that are closest to the expert trajectory in different `fetch_pick_and_place` tasks (orange) overlayed over a kernel density estimate of the goal occupancy in the full training dataset.

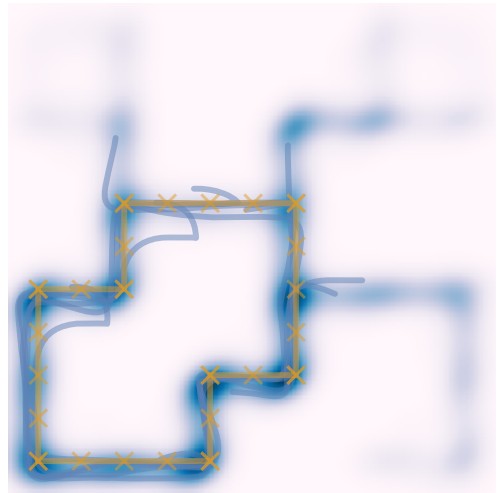
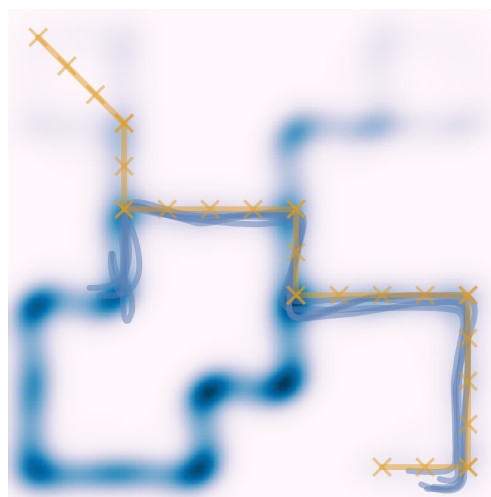

(a) `circle-dense`    (b) `path-dense`

*Figure 20.* The 5 trajectories (blue) from the dataset that are closest to the expert trajectory in different `pointmaze_medium` tasks (orange) overlayed over a kernel density estimate of the goal occupancy in the full training dataset.

## C.9. Hyperparameters

*Table 7.* TD-MPC2 Hyperparameters. We have adopted these unchanged from Hansen et al. (2024)

| Name | Value |
|---|---|
| `lr` | 3e-4 |
| `batch_size` | 256 |
| `n_steps` ("horizon") | 3 |
| `rho` | 0.5 |
| `grad_clip_norm` | 20 |
| `enc_lr_scale` | 0.3 |
| `value_coef` | 0.1 |
| `reward_coef` | 0.1 |
| `consistency_coef` | 20 |
| `tau` | 0.01 |
| `log_std_min` | -10 |
| `log_std_max` | 2 |
| `entropy_coef` | 1e-4 |

| Name | Value |
|---|---|
| `num_bins` | 101 |
| `vmin` | -10 |
| `vmax` | 10 |
| `num_enc_layers` | 2 |
| `enc_dim` | 256 |
| `num_channels` | 32 |
| `mlp_dim` | 512 |
| `latent_dim` | 512 |
| `bin_dim` | 12 |
| `num_q` | 5 |
| `dropout` | 0.01 |
| `simnorm_dim` | 8 |

*Table 8.* Hyperparameters used for iCEM (Pinneri et al., 2020). We use the implementation from Pineda et al. (2021).

(a) ICEM hyperparameters for all MPC planners.

| Name | Value |
|---|---|
| num_iterations | 4 |
| population_size | 512 |
| elite_ratio | 0.01 |
| population_decay_factor | 1.0 |
| colored_noise_exponent | 2.0 |
| keep_elite_frac | 1.0 |
| alpha | 0.1 |

(b) ICEM hyperparameters for curious exploration.

| Name | Value |
|---|---|
| num_iterations | 3 |
| population_size | 512 |
| elite_ratio | 0.02 |
| population_decay_factor | 0.5 |
| colored_noise_exponent | 2.0 |
| keep_elite_frac | 1.0 |
| alpha | 0.1 |
| horizon | 20 |

## C.10. FB Implementation Details

Since there is no implementation available for FB-IL directly, we have adopted the code for FB (Touati & Ollivier, 2021) according to the architectural details in appendix D.3 and the hyperparameters in appendix D.4 of FB-IL (Pirotta et al., 2024). The main architectural changes consisted of changing the state input of the $B$ networks to only a goal input, as suggested in (Touati & Ollivier, 2021) as well as adding a last layer in the $B$ networks for L2 projection, batch normalization, or nothing, depending on the environment.

We follow the specifications of Pirotta et al. (2024) whenever possible. As halfcheetah and maze are also used in their evaluations we have adopted their hyperparameters for these environments. For our fetch environments, we used the hyperparameters most common in the environments except for the discount $\gamma$ which we adjusted to 0.95 to account for the shorter episode length. Finally, we have found that the FB framework seems to be ill-adjusted to be trained on an order of magnitude less data then in the original experiments (Touati & Ollivier, 2021; Pirotta et al., 2024). For some environments, performance started to deteriorate rather quickly, so we to report the best performance encountered during training when evaluating every 50k steps (see D.2). We provide the full set of hyperparameters in table 9.

*Table 9.* Hyperparameters used for FB-IL training. Closely follows table 1 in appendix D.4 of (Pirotta et al., 2024) for halfcheetah and maze.

| Environment | fetch | halfcheetah | maze |
|---|---|---|---|
| Representation dimension | 50 | 50 | 100 |
| Batch size | 2048 | 2048 | 1024 |
| Discount factor $\gamma$ | 0.95 | 0.98 | 0.99 |
| Optimizer | Adam | Adam | Adam |
| learning rate of $F$ | $10^{-4}$ | $10^{-4}$ | $10^{-4}$ |
| learning rate of $B$ | $10^{-4}$ | $10^{-4}$ | $10^{-6}$ |
| learning rate of $\pi$ | $10^{-4}$ | $10^{-4}$ | $10^{-6}$ |
| Normalization of $B$ | L2 | None | Batchnorm |
| Momentum for target networks | 0.99 | 0.99 | 0.99 |
| Stddev for policy smoothing | 0.2 | 0.2 | 0.2 |
| Truncation level for policy smoothing | 0.3 | 0.3 | 0.3 |
| Regularization weight for orthonormality | 1 | 1 | 1 |
| Numer of training steps | $2 \cdot 10^6$ | $2 \cdot 10^6$ | $2 \cdot 10^6$ |

# D. Hyperparameter Searches

## D.1. Classifier Threshold

As mentioned in the main text, we perform an extensive hyperparameter search for the threshold value of the goal classifier (Cls) for the myopic methods Pi+Cls and MPC+Cls as well as for the ablation of our method ZILOT+Cls. In figures 22 and 21 we show the performance of the three respective planners in all five environments and denote the threshold values that yield the best performance per environment. Interestingly, in some of the `fetch` environments not all tasks attain maximum performance with the same threshold value showing that this hyperparameter is rather hard to tune.

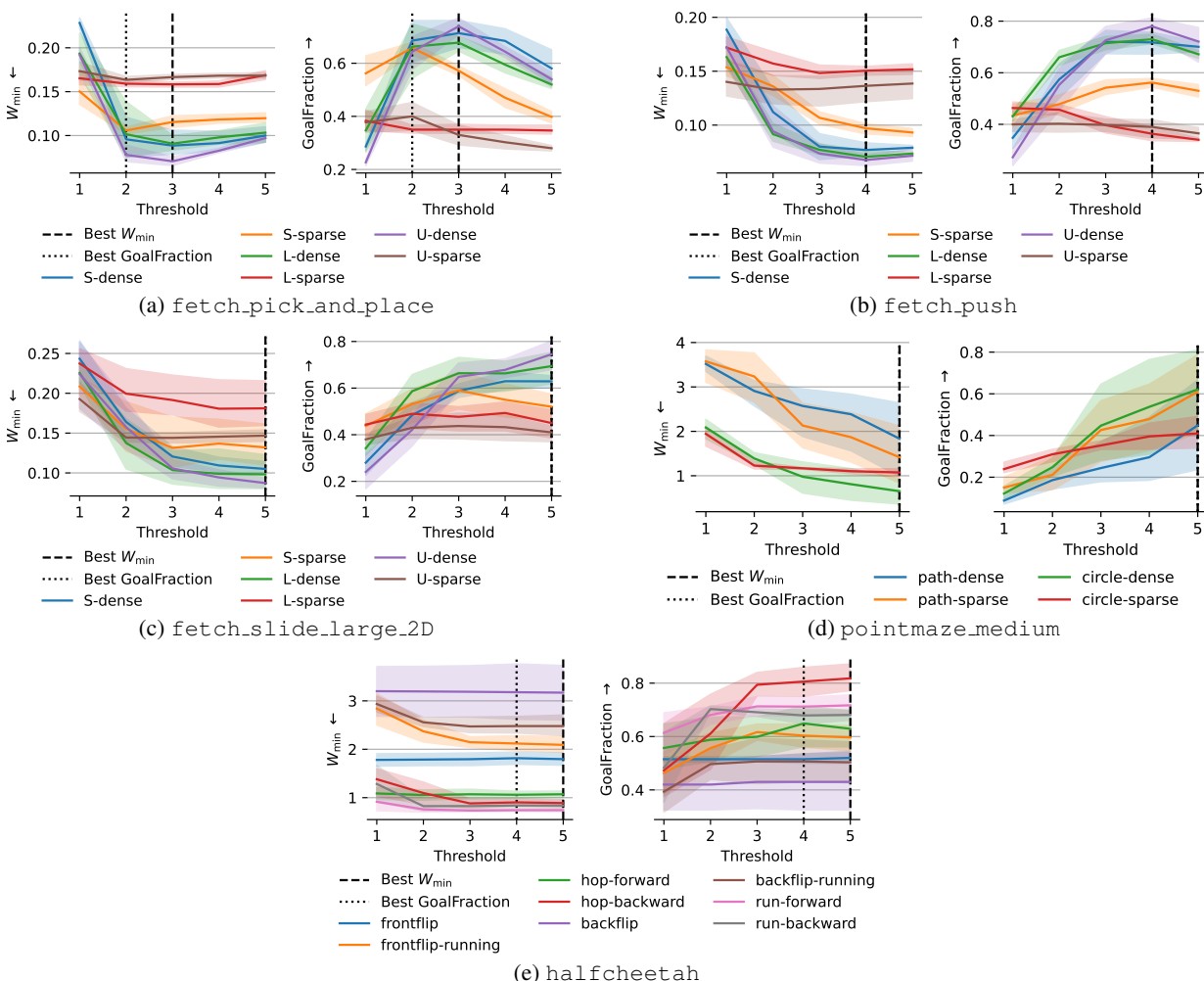

*Figure 21.* ZILOT+Cls hyperparameter search for Cls threshold.

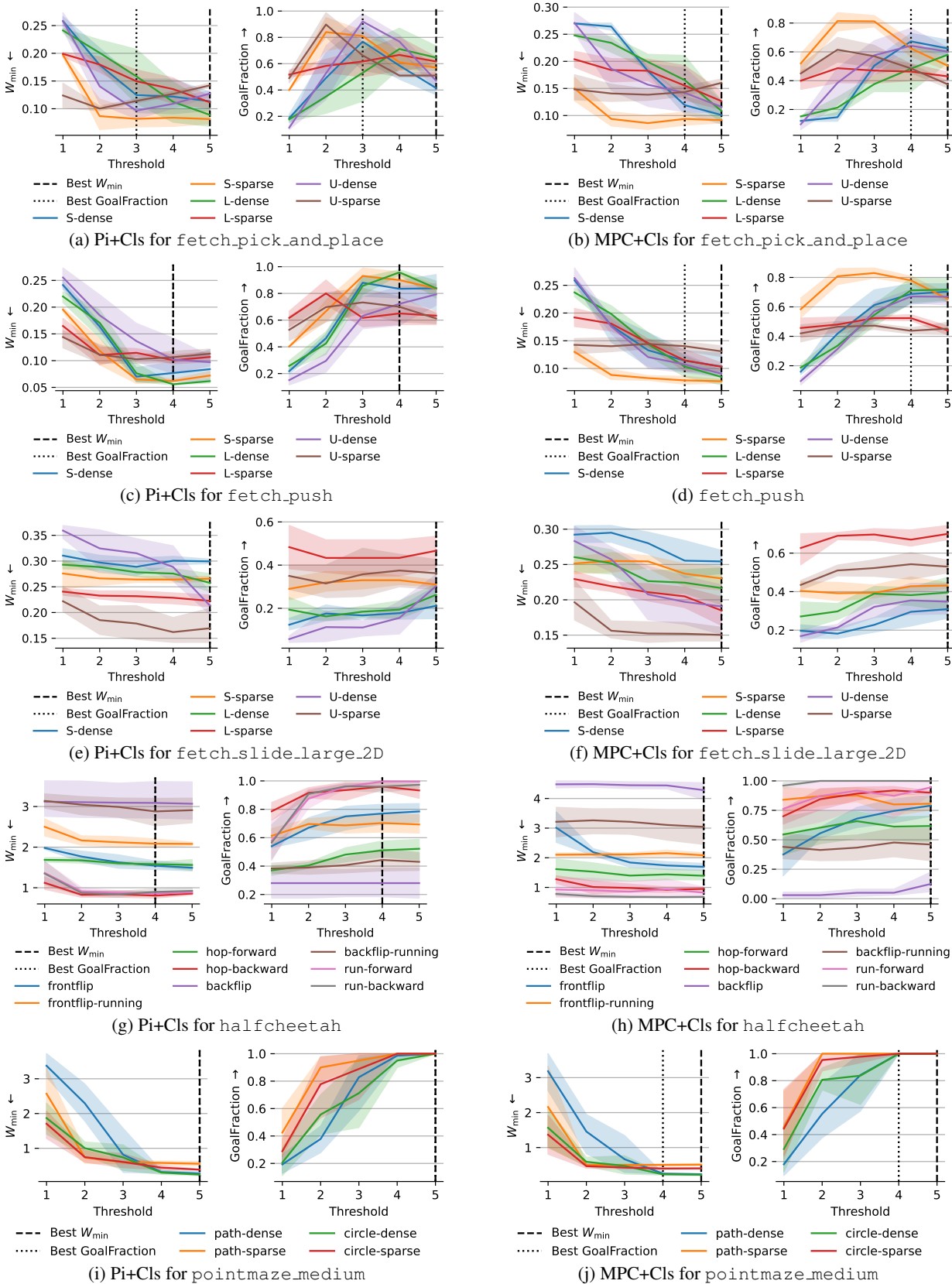

*Figure 22.* Pi+Cls and MPC+Cls hyperparameter searches for Cls threshold in each environment.

## D.2. FB-IL Training steps.

As mentioned in section C.10 we report the best evaluation results of all FB methods that occur during training. In figures 23 and 24 we report the evaluation performance for different training lengths.

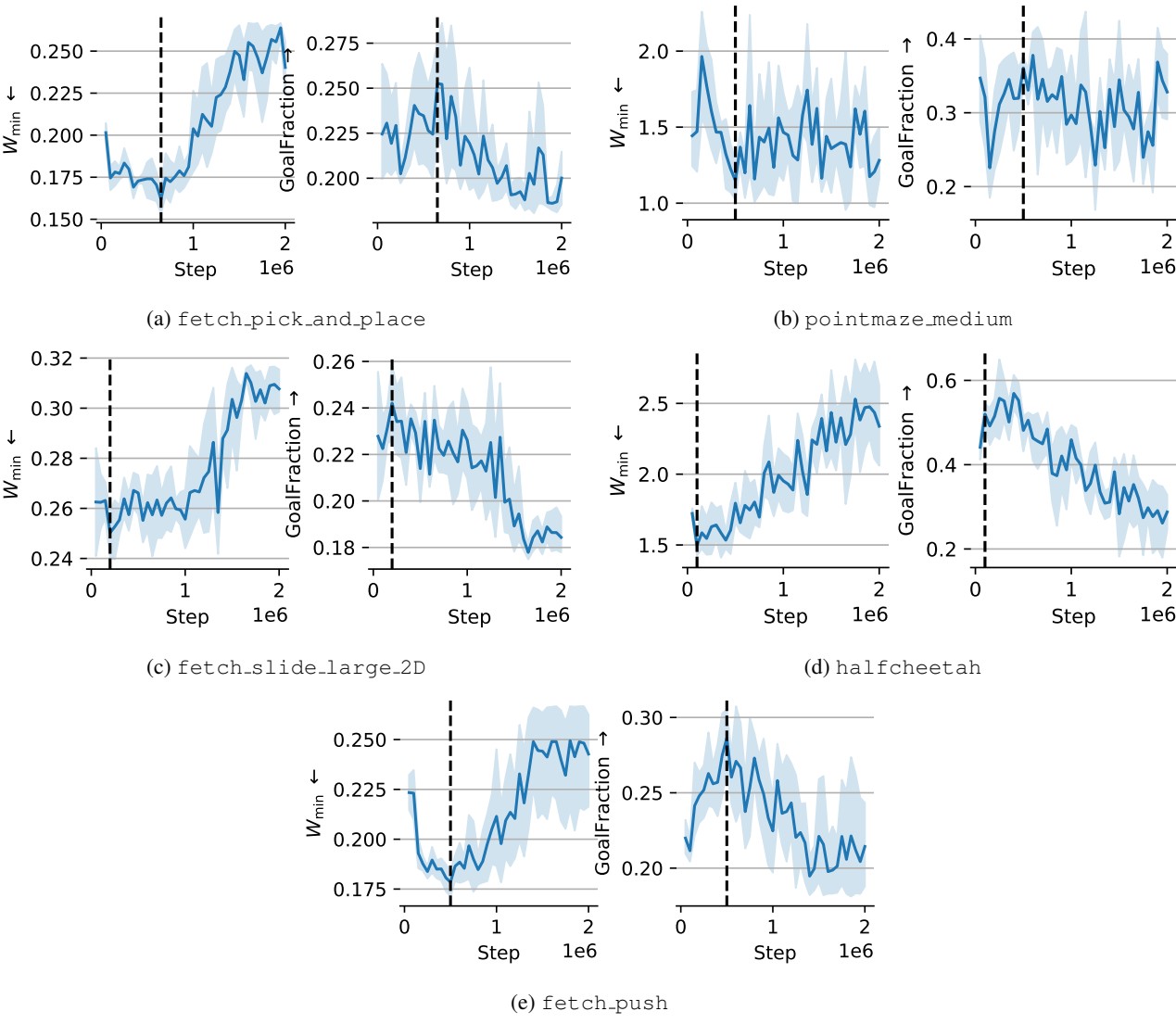

*Figure 23.* $ER_{FB}$ evaluation performance during training.

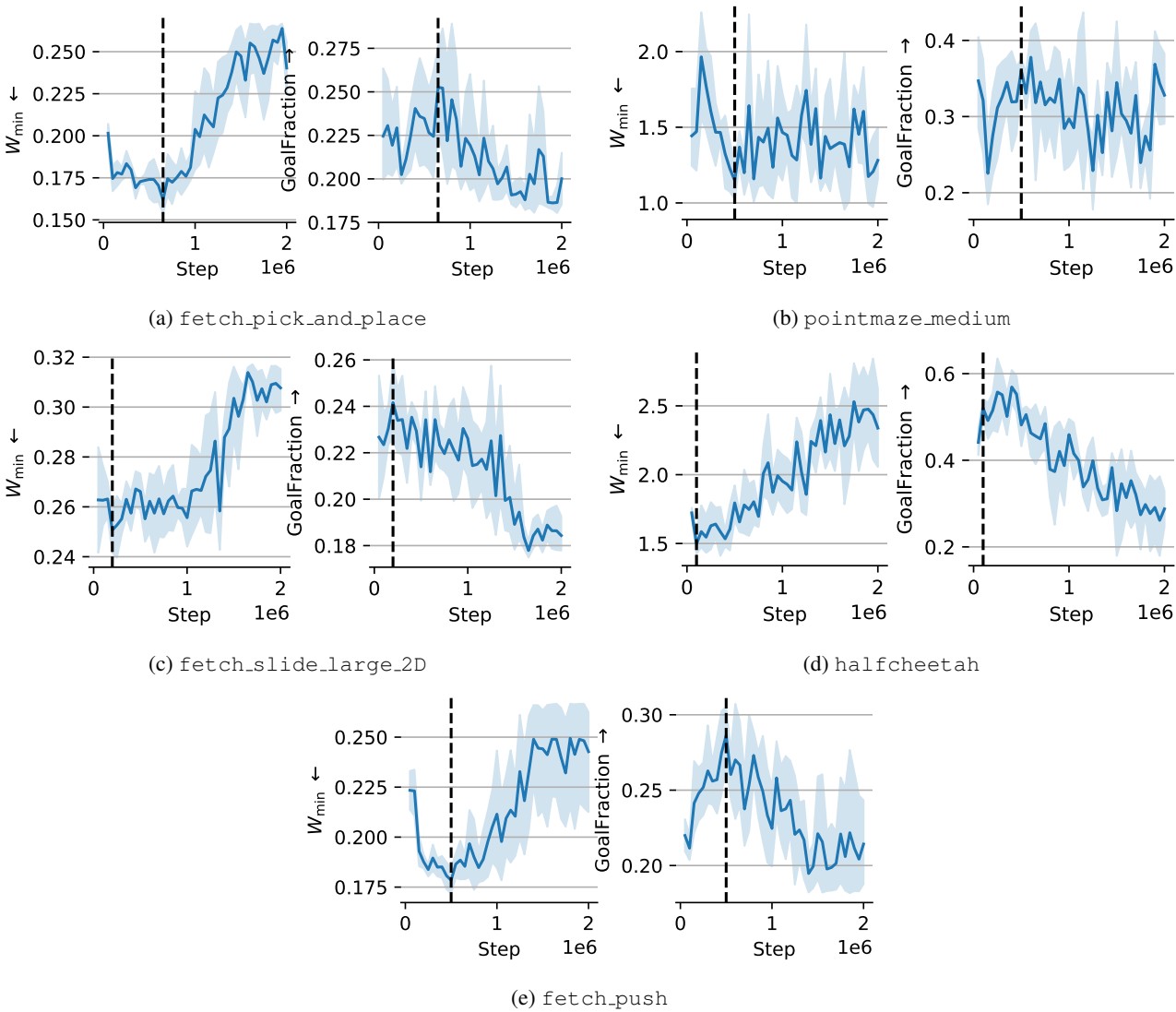

(a) fetch_pick_and_place

(b) pointmaze_medium

(c) fetch_slide_large_2D

(d) halfcheetah

(e) fetch_push

*Figure 24.* RER_FB evaluation performance during training.

# E. Additional Qualitative Results

In the following, we present all goal-space trajectories across all planners, tasks, and seeds presented in this work. Note that since the tasks of the fetch environments display some natural symmetries, we decided to split evaluations between all four symmetrical versions of them. Further, we quickly want to stress that these trajectories are shown in goal-space. This means that if the cube in fetch is not touched, as is the case in some cases for ZILOT+$h$, then the trajectory essentially becomes a single dot at the starting position. Also note that Pi+Cls is completely deterministic, which is why its visualization appears to have less trajectories.

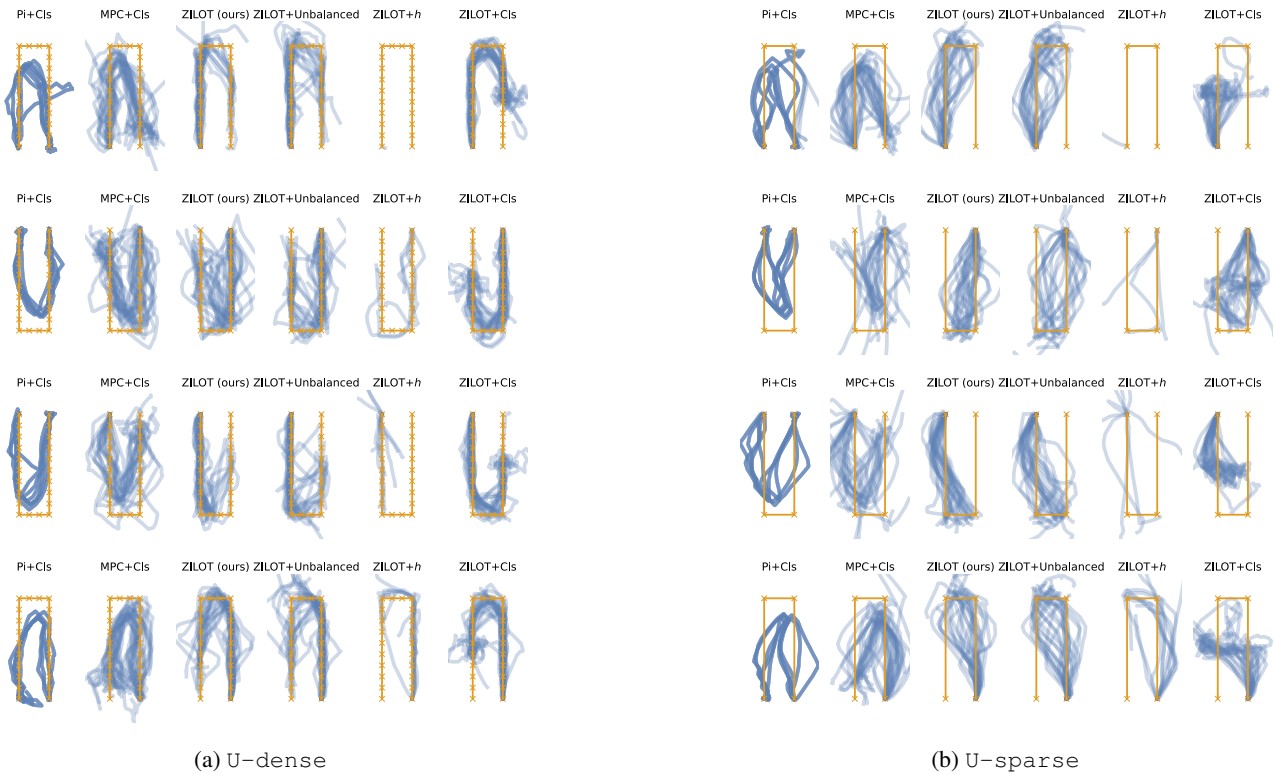

(a) U-dense

(b) U-sparse

*Figure 25.* fetch_pick_and_place

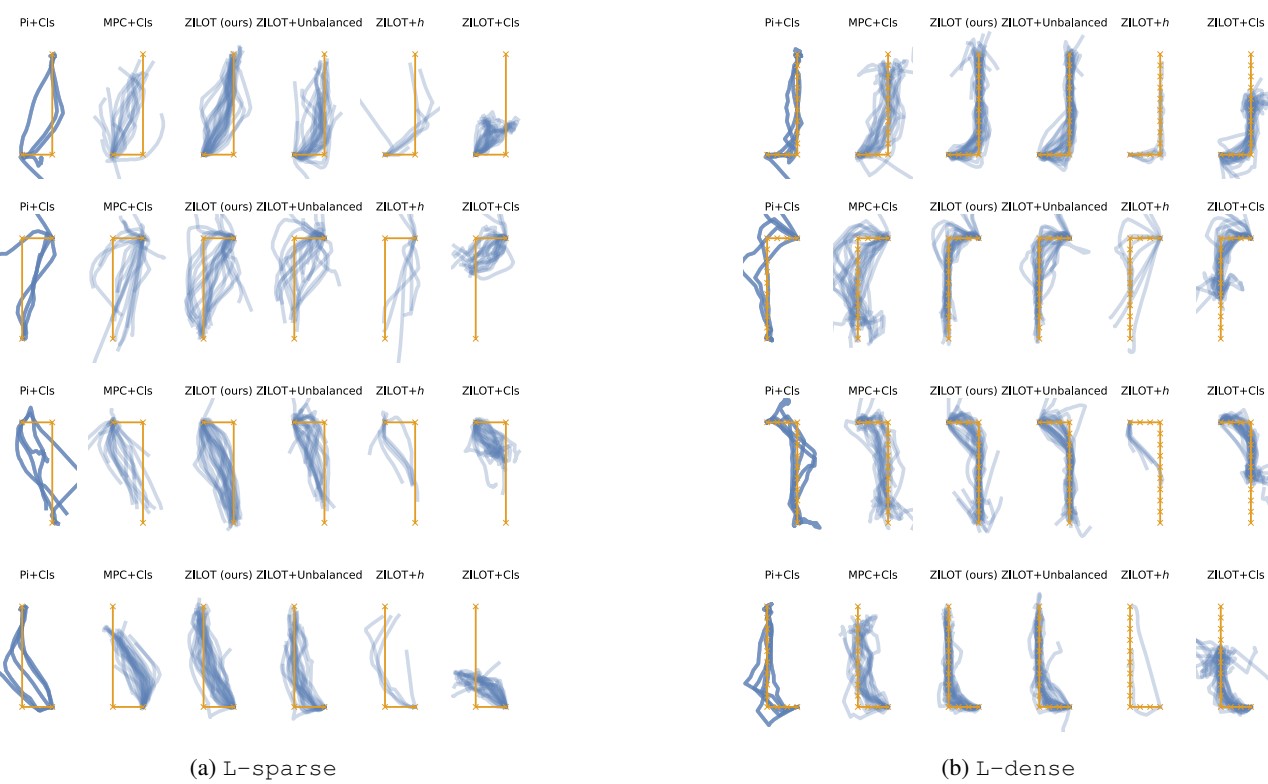

(a) L-sparse

(b) L-dense

*Figure 26.* fetch_pick_and_place

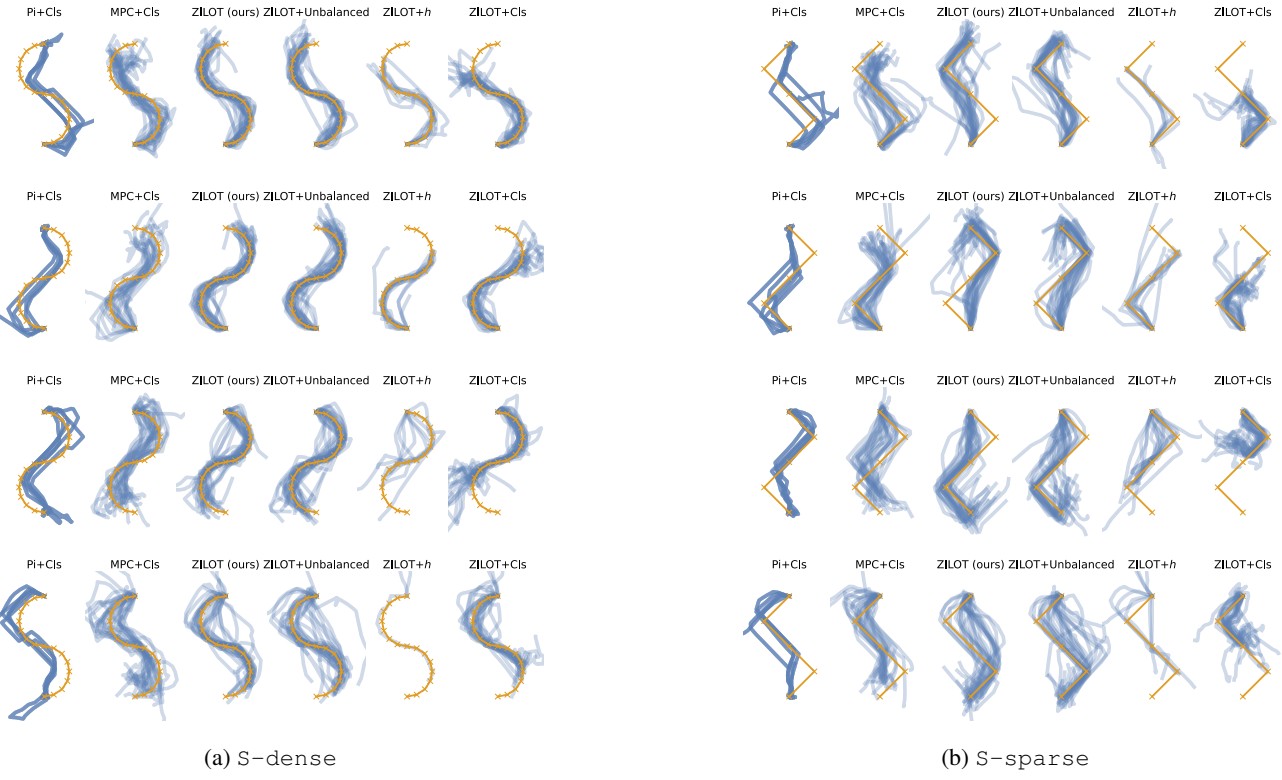

(a) S-dense

(b) S-sparse

*Figure 27.* fetch_pick_and_place

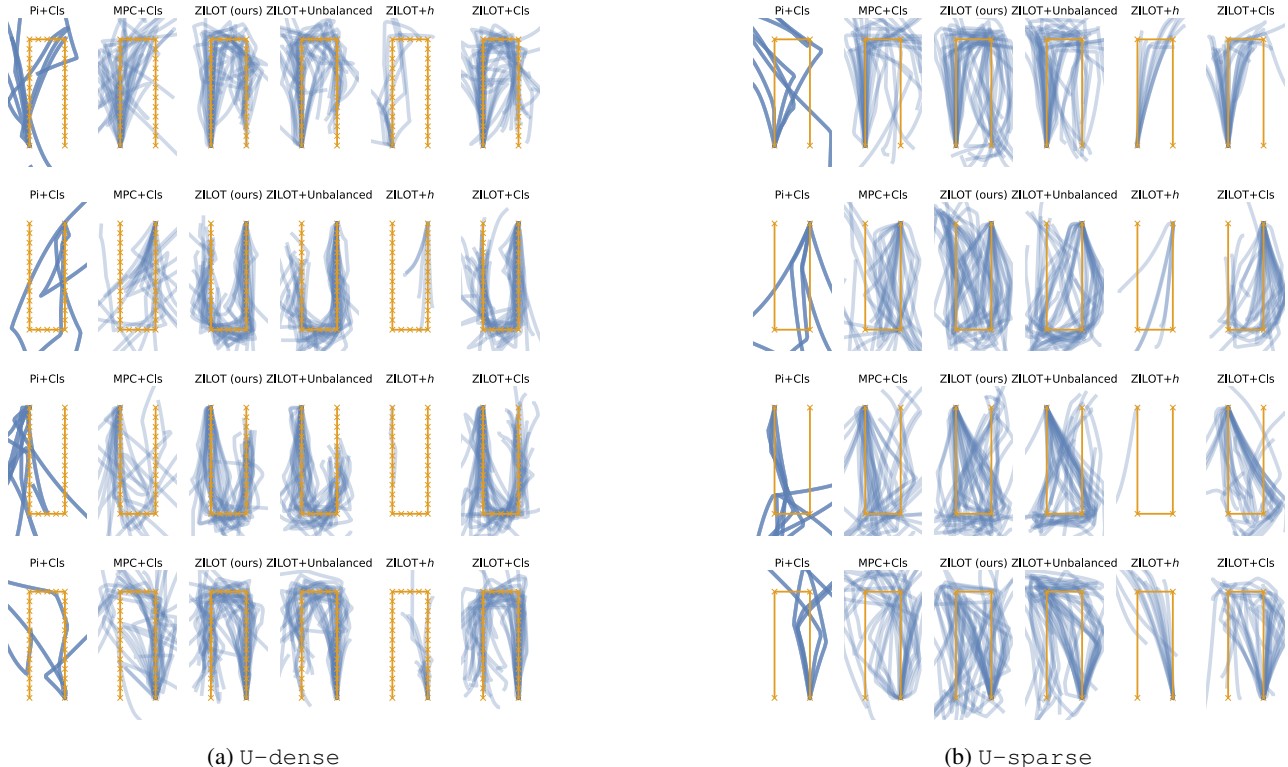

(a) U-dense                    (b) U-sparse

*Figure 28.* fetch_slide_large_2D

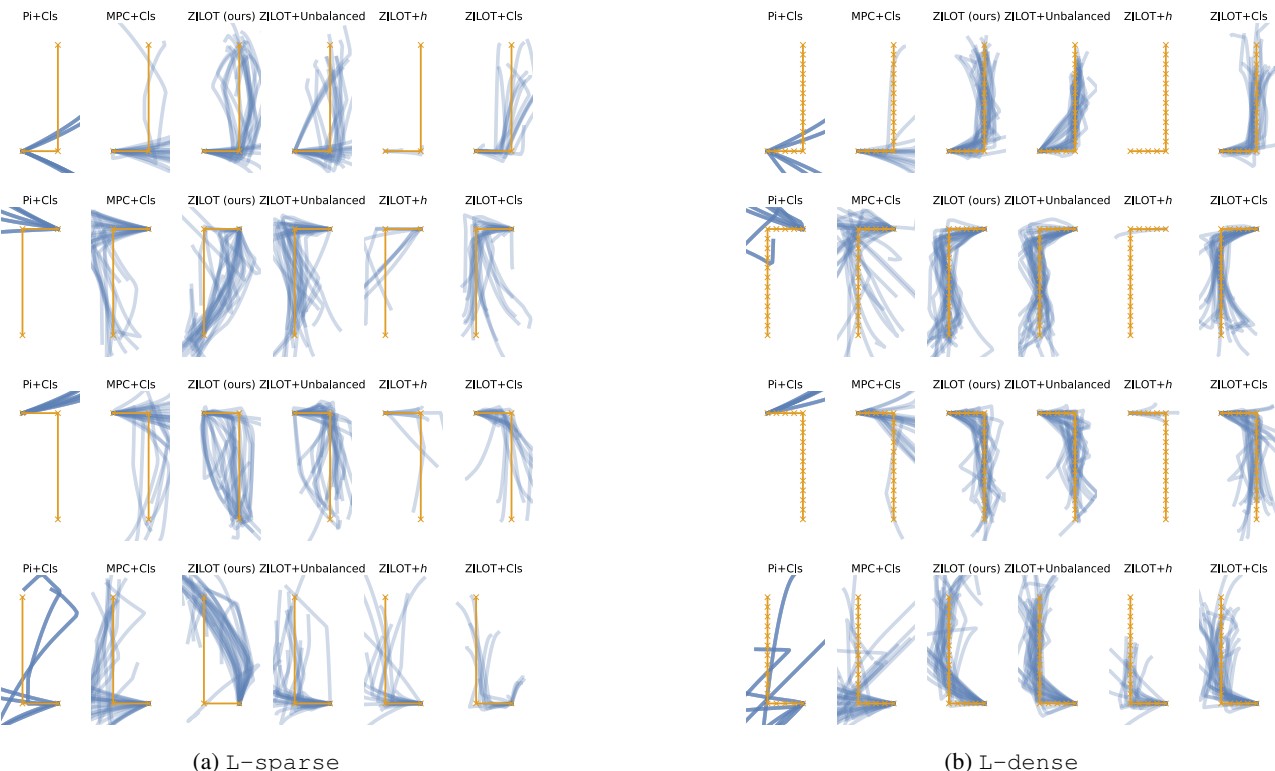

(a) L-sparse                    (b) L-dense

*Figure 29.* fetch_slide_large_2D

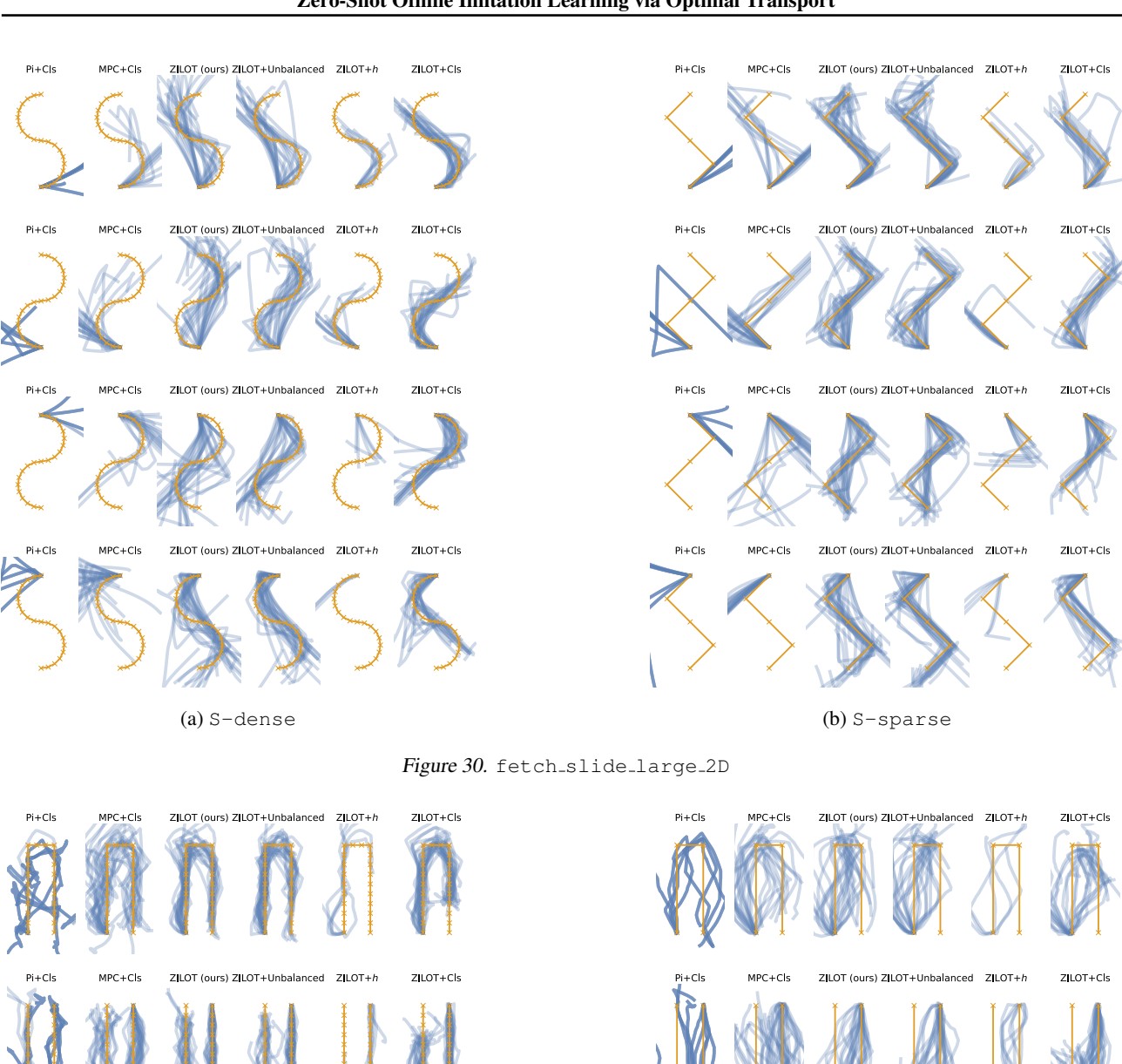

(a) `S-dense`

(b) `S-sparse`

*Figure 30.* `fetch_slide_large_2D`

(a) `U-dense`

(b) `U-sparse`

*Figure 31.* `fetch_push`

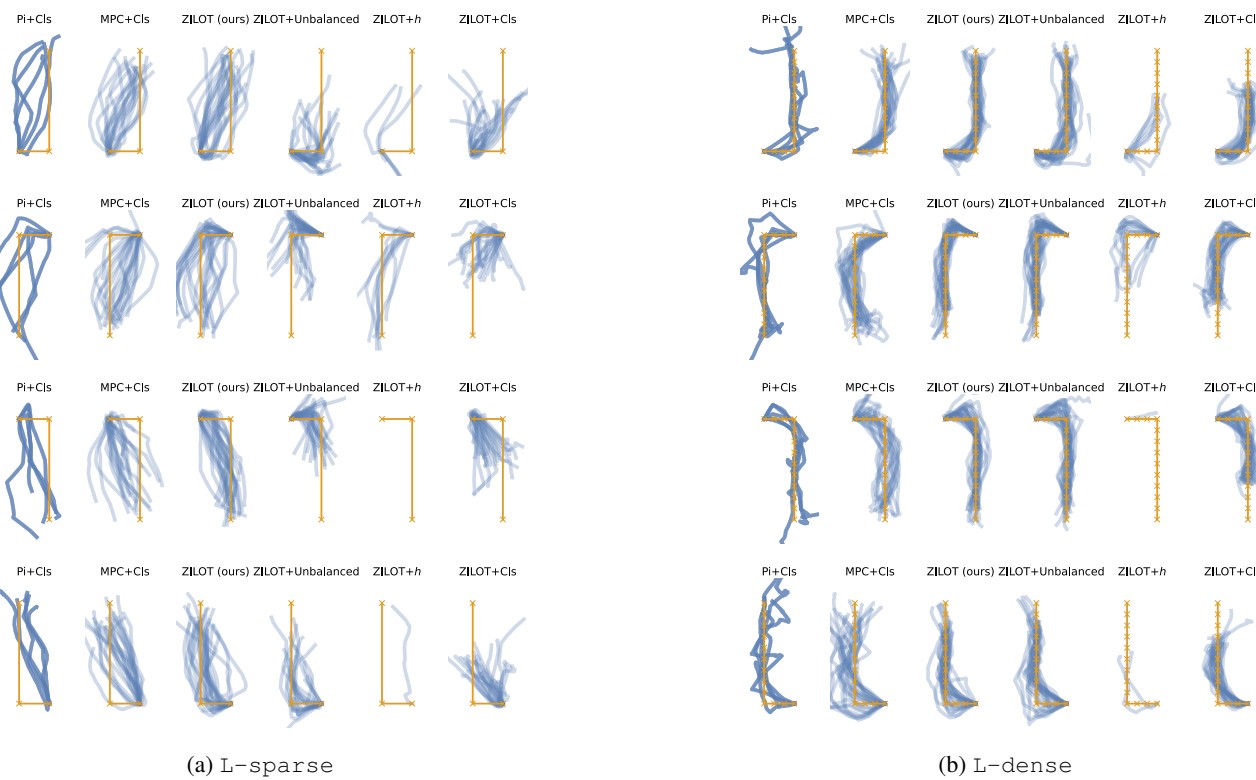

(a) L-sparse

(b) L-dense

*Figure 32.* fetch_push

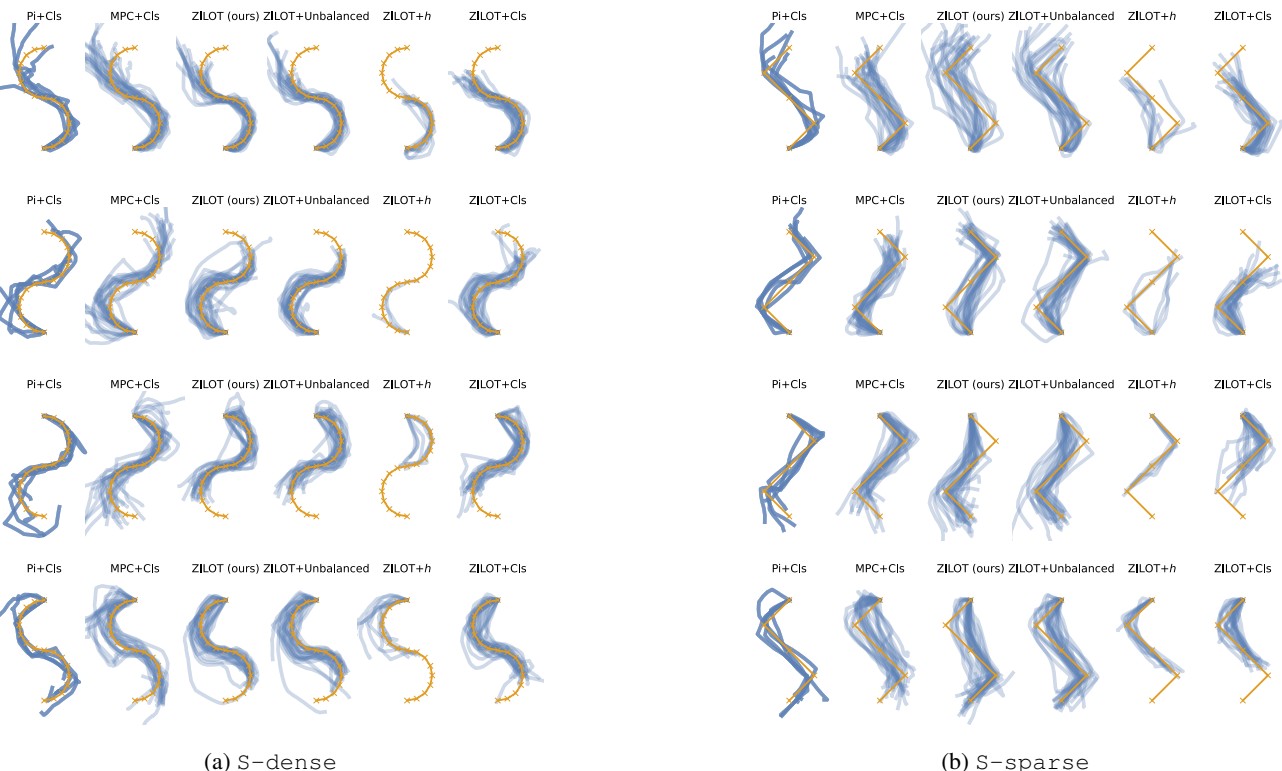

(a) S-dense

(b) S-sparse

*Figure 33.* fetch_push

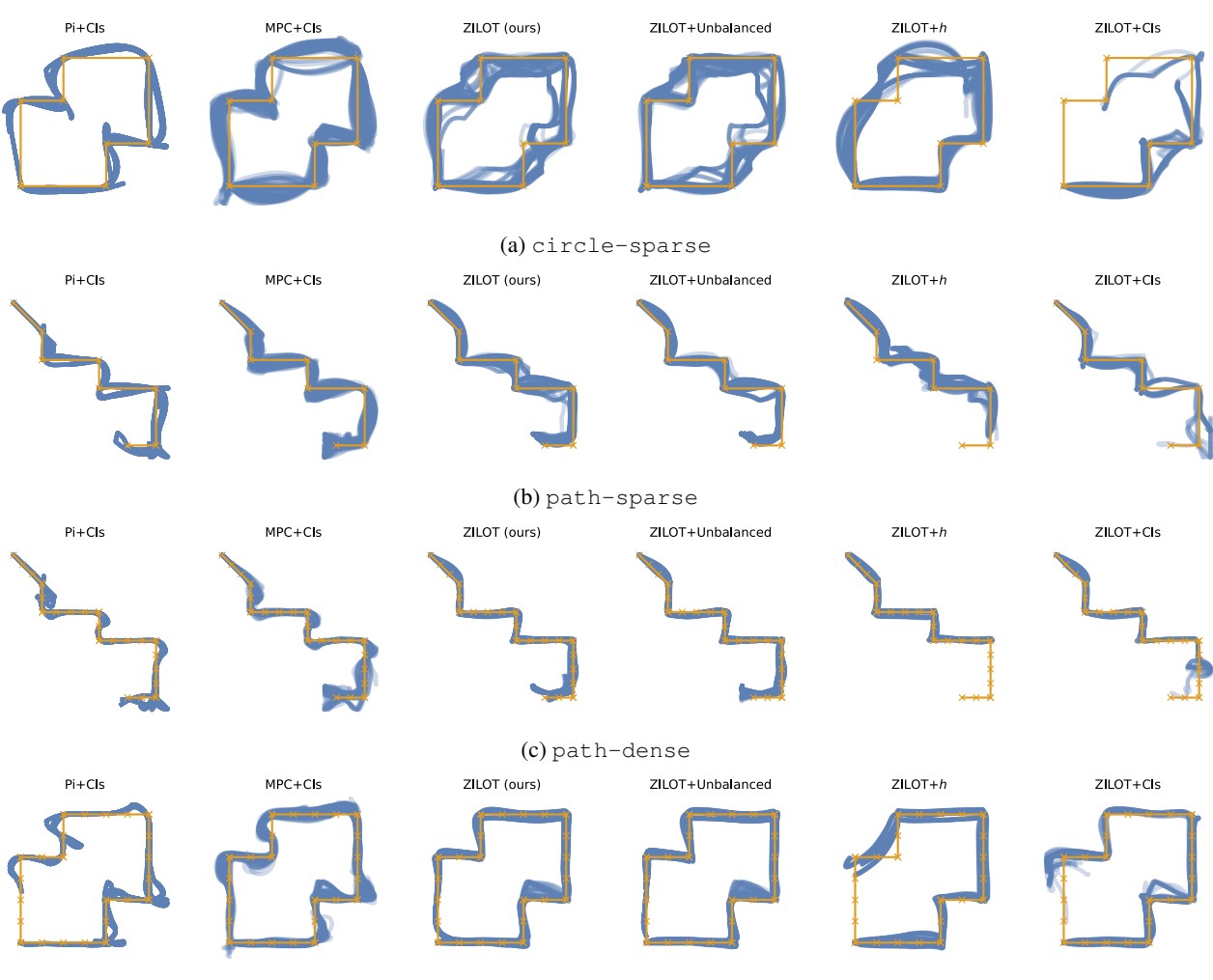

(a) `circle-sparse`

(b) `path-sparse`

(c) `path-dense`

(d) `circle-dense`

*Figure 34.* `pointmaze_medium`

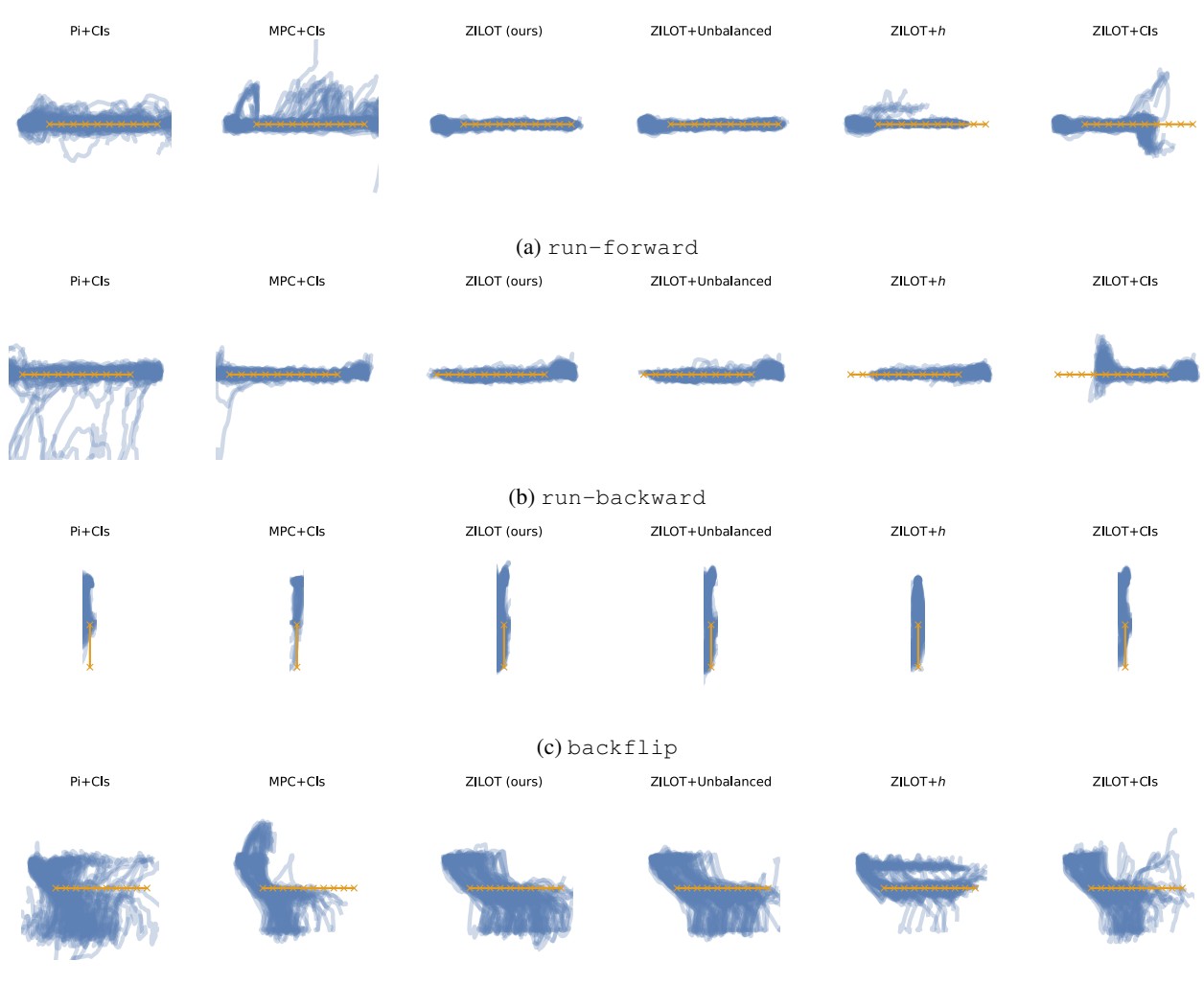

(a) run-forward

(b) run-backward

(c) backflip

(d) hop-forward

*Figure 35.* halfcheetah part 1

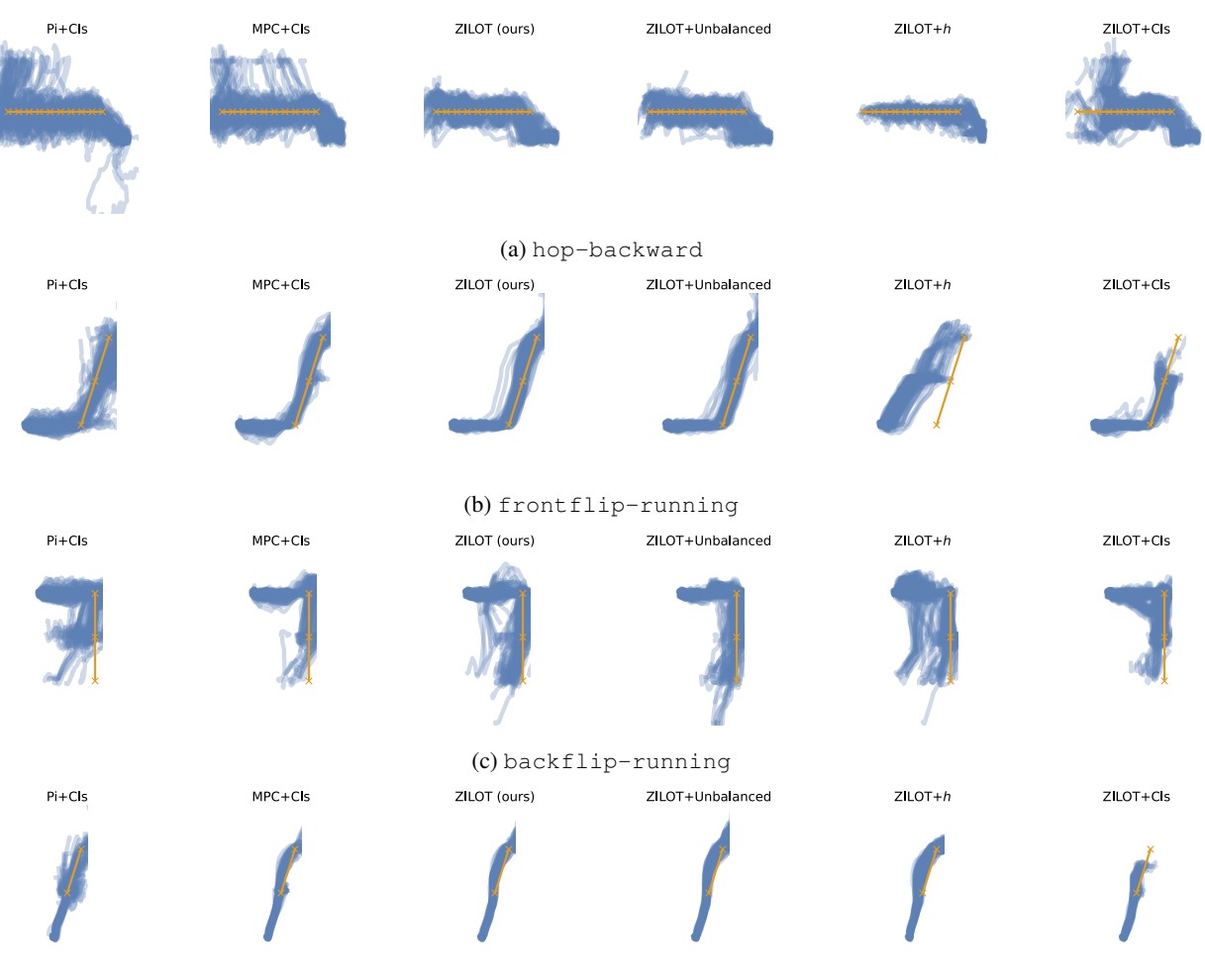

(a) `hop-backward`

(b) `frontflip-running`

(c) `backflip-running`

(d) `frontflip`

*Figure 36.* `halfcheetah` part 2

