# OpenReview forum: "Zero-Shot Offline Imitation Learning via Optimal Transport"
_ICML.cc/2025/Conference — ICML 2025 poster_

### Official Review · Reviewer_H8vF · 2025-02-23

**Overall Recommendation:** 3

**Summary:**

This paper proposes ZILOT, a MPC-style inference-time trajectory optimization technique that can learn from a single incomplete state-only expert demonstration with pretraining of a dynamic model from an unlabeled state-action dataset. The paper theoretically proves prior method, which is using a goal recognizer with a goal-conditioned policy, will encounter myopic issues and is suboptimal. This paper instead uses MPC, with the Sinkhorn distance between state occupancies of the rollout policy and expert demonstrations as the cost optimization objective. The distance metric between two states are the expected steps from one state to another, which is learned through a goal-conditioned value function. The optimal transport between occupancies are discretized into matching between the states. To determine whether expert states are reachable, another value function is learned to provide an estimate. On several testbeds, the proposed method outperforms existing baselines.

## Update After Rebuttal

I have updated my score accordingly during the author-discussion period; no further update.

**Claims And Evidence:**

Overall, I feel the claim "zero-shot" is somewhat strange to me; I think this is one of my major concern for this paper. In this paper, the definition of "zero-shot" seems to be "(a single) demonstration devoid of actions (which is rough and partial), such as specifying only a few checkpoints" (Sec. 1).

However:

1. in one of the zero-shot prior work the authors referred to in Sec. 1, Pathak et al. [1], the definition of zero-shot is "the agent never has access to expert actions during training or for the task demonstration at inference" (as specified in the abstract). This is, however, not the case in this paper; $\mathcal{D}\_\beta$ contains actions as specified in Sec. 2.1.

2. In another paper the authors referred to in Sec. 1, Pirotta et al. [2], the author writes in their Sec. 1: "1) ... no prior knowledge or demonstrations of the behavior to be imitated are available, and only a dataset of unsupervised trajectories is provided (which is the case for this paper); 2) ... solve any imitation task without any additional samples on top of the demonstrations, and without solving any complex RL problem (which means offline, but not necessarily without actions) ... computation needed to return the imitation policy should be minimal (which is questionable for this work)".

3. in "zero-shot" reinforcement learning [3], which Pirotta et al. [2] refers to (as Pirotta et al. did not formally define the word "zero-shot"), the definition of zero-shot is "solve any RL task in a given environment, instantly with no additional planning or learning, after an initial reward-free learning phase", which is also not satisifed by this paper.

4. The authors also use LLM's in-context learning ability as an example for zero-shot capabilities in the beginning of this paper. However, LLM's in-context learning ability is "no any training related to the problem", which is different from this paper with a world model training stage.

5. according to the author's definition in this paper (state-action offline dataset + state-only incomplete / goal-only trajectories), there are many other papers that fits into this definition in the name of "learning from observations", such as IQLearn [4], TAILO [5], PW-DICE [6], SMODICE [7], SAIL [8], RCE [9] (which is goal-based IL), AILOT [10]. The authors seem to only have cited SMODICE in the appendix as an example of forward-backward framework.

Therefore, I feel the word "zero-shot" is not well-defined in this paper.


**Reference**

[1] D. Pathak et al. Zero-Shot Visual Imitation. In ICLR, 2018.

[2] M. Pirotta et al. Fast Imitation via Behavior Foundation Models. In ICLR, 2024.

[3] A. Touati et al. Does Zero-Shot Reinforcement Learning Exist? In 34th Offline Reinforcement Learning Workshop at Neural Information Processing Systems, 2022.

[4] D. Garg et al. IQ-Learn: Inverse soft-Q Learning for Imitation. In NeurIPS, 2021.

[5] K. Yan et al. A Simple Solution for Offline Imitation from Observations and Examples with Possibly Incomplete Trajectories. In NeurIPS, 2023.

[6] K. Yan et al. Offline Imitation from Observation via Primal Wasserstein State Occupancy Matching. In ICML, 2024.

[7] Y. J. Ma et al. Versatile Offline Imitation from Observations and Examples via Regularized State-Occupancy Matching. In ICML, 2022.

[8] F. Liu et al. State Alignment-based Imitation Learning. In ICLR, 2020.

[9] B. Eysenbach et al. Replacing Rewards with Examples: Example-Based Policy Search via Recursive Classification. In NeurIPS, 2021.

[10] M. Borbin et al. Align Your Intents: Offline Imitation Learning via Optimal Transport. In ICLR, 2025.

**Essential References Not Discussed:**

I suggest the authors check "learning from observations" in the imitation learning literature, as mentioned in "Claims and Evidence" point 5. Currently, I feel multiple papers are missing from the literature discussion. Also, I would suggest the authors to discuss the relation of this work with the ICVF paper [1], which also uses a goal-conditioned value function and aims to learn dynamics through value function from action-free data, which is similar to the goal-conditioned valuer function in this paper.

**References**

[1] D. Ghosh et al. Reinforcement Learning from Passive Data via Latent Intentions. In ICML, 2023.

**Experimental Designs Or Analyses:**

Yes, I think the experiment are generally solid as many details as well as visualizations are given in the appendix. The results seem reasonable which shows the proposed method outperforms several baselines, and ablations are made in Sec. 5.3. Many environment variants are tested in Tab. 2, which I feel sufficient to prove the performance of the proposed method.

**Methods And Evaluation Criteria:**

Yes, the proposed methods and evaluation criteria over make sense for the problem. There are a few possible concerns and improvements though:

**Concerns**

In the paper, the authors try to estimate the state occupancy using $\rho^\pi_N\approx\frac{1}{N+1}\sum\_{t=0}^N\delta\_{s\_t}$. While empirically, works such as OTR [1] mentioned in the paper have proved the success of the discretization, it is worth noting that the estimation of optimal transport between distributions from discrete samples are very inaccurate [2, 3]; in fact, one will need exponentially many samples with respect to the number of state dimensions to accurately estimate the Wasserstein distance between distribution. Thus, the story of approximation from Eq. 12 to the end of "occupancy estimation" does not sound convincing enough.

**Improvements**

1. The paper did not specify how the "world model" is trained from the offline dataset, which I feel is an important part of the method (as the authors indicated in Appendix C.4, it seems to follow prior work, but the paper should still introduce the method and its implementation in this paper to be self-contained). Moreover, the paper did not specify what a goal abstraction $\phi(\cdot)$ is and how it is implemented in the proposed method or baselines (given this, it is unexplained on the objective right above Eq. 18 why $W(\phi(s);\phi(s'))$ has goal abstraction on both state inputs but $V(s,\phi(s'))$ only has goal abstraction on the second state input. This is another major concern of mine.

2. It will be better if the authors can show the result of ZILOT on more environments (walker2d, ant, antmaze, franka kitchen etc.), which are usually accompanieed with the halfcheetah environment in the literature.

**References**

[1] Y. Luo et al. Optimal Transport for Offline Imitation Learning. arXiv: In ICLR, 2023.

[2] J. Stanczuk et al. Wasserstein GANs Work Because They Fail (to Approximate the Wasserstein Distance). arXiv:2103.01678.

[3] J. Weed et al. Sharp Asymptotic and Finite-Sample Rates of Convergence of Empirical Measures in Wasserstein Distance. arXiv:1707.00087.

**Other Comments Or Suggestions:**

See the other parts of the review.

**Other Strengths And Weaknesses:**

**Other Strength**

Overall, I feel the high-level idea of this paper is clearly conveyed and easy to follow. The shortcoming of the prior methods are clearly illustrated in Fig. 1 and rigorously stated afterwards in Sec. 3.

**Other Weakness**

The method needs to solve optimization problem for every step, which is inefficient; as the authors suggested in Appendix C.3, the method only runs at 0.5 to 3Hz, which is too slow for practical uses (and even slower for simulations).

**Questions For Authors:**

I have two question for the authors:

1. Is the proposed method robust against dynamic mismatch between expert demonstration and the agent? I would expect the method to be robust if it can work with incomplete state-only trajectories and uses Wasserstein distance to match those states. An example of such experiment can be found in Appendix H of the SMODICE [1] paper.

**References**

[1]  Y. J. Ma et al. Versatile Offline Imitation from Observations and Examples via Regularized State-Occupancy Matching. In ICML, 2022.

**Relation To Broader Scientific Literature:**

This paper is beneficial for the Reinforcement Learning (RL) / Imitation Learning (IL) community and robotics community. It does not have significant impact for researchers out of these communities.

**Theoretical Claims:**

The paper has one theoretical claim in Sec. 3, which is about the suboptimality of the method proposed in Pathak et al. [1]. The authors claim that there exists some controllable Markov chain and sequence of goals that the existing method, even with optimal solution, cannot all achieve. The authors give a simple counterexample as a proof and it seems correct to me. To make it more rigorous, I would suggest add "almost surely" or "with probability 1" to the conclusion "will not reach all goals in the sequence".

**Reference**

[1] D. Pathak et al. Zero-Shot Visual Imitation. In ICLR, 2018.

---

> ### Author Rebuttal · Authors · 2025-03-31
>
> # Definition of zero-shot
> We would like to clarify our definition of “zero-shot”. _We refer to a method as “zero-shot” if it retrieves an optimal policy for unseen objectives provided at test-time, with modest compute overhead_. “Zero-shot” methods may be allowed a compute-heavy pre-training phase, which should not be informed of the downstream task. This definition aligns with [1], which is thus a method for zero-shot IL. Similarly, [2] and [3] perform zero-shot IL and RL, respectively. IQLearn, TAILO, PW-DICE, SMODICE, AILOT, and the majority of offline IL methods, would not qualify as zero-shot IL methods, as they need the objective (in the case of IL, the demonstration) to be available during pre-training.
>
> Existing definitions do not specify what constitutes a “modest compute overhead”. In [1], “zero-shot” methods require <5 minutes to imitate a single trajectory, while non-zero-shot methods require >3 hours (fig. 2 in [1]). ZILOT takes <4 minutes to imitate a trajectory in our tasks, which is comparable to more involved FB-IL variants, and orders of magnitude faster than non-zero-shot methods, e.g., BC. We would thus argue that ZILOT performs zero-shot IL.
>
> This definition of “zero-shot” depends on when demonstrations are provided, not on the information they contain. They may contain actions (as for FB-IL-BC), or may not. We can qualify this distinction: we _may define a method to be “learning from observations” if it imitates demonstrations that do not contain actions_. ZILOT and [2] would then belong to this category. For both, an action-labeled offline dataset (not demonstrations, but arbitrary trajectories) is provided during pre-training to convey action semantics.
>
> We thank the reviewer for encouraging this discussion, which we hope clarifies our definitions. We will update our submission accordingly.
>
> # Concerns - Finite sample approximation of OT
> We acknowledge that our occupancy matching objective is sample-based and thus subject to approximation errors. As mentioned, for both OTR and our work, this approach performs well empirically. We hypothesize that this might be due to the adoption of entropy regularization [4], or to a low dimensionality (as defined in [5]) of occupancy under smooth dynamics in discrete time.
>
> # Suggested Improvements
> ## World Model Training
> We will expand Appendix C.2 with training objectives for our practical choice of world model (TD-MPC2), and refer to it prominently in the main text.
>
> ## Goal Abstraction
> We introduce the general concept of a goal-abstraction in Sec. 2.1. and specify the exact functions used for each environment we evaluate in Table 5 in the Appendix (e.g., for Fetch we use the cube pose). As is standard in GC-RL the goal-abstraction is known [6].
> ZILOT’s planning objective (Eq. 17) matches the sampled state occupancy to the expert goal occupancy, and thus uses a standard GCVF $V(s, g)$.
>
> Eq. 18 is used for estimating the time the expert required for the demonstration; as the demonstration only contains goals, it requires a separate value function estimating distances between pairs of goals ($W(g, g’)$). It is only used as a heuristic for selecting a part of the expert trajectory to match at each step, which is necessary for finite-horizon optimization.
>
> ## Extra Evaluation Environments
> Please refer to our response to Reviewer LRC1, where we motivate our choice of environments and provide evaluations of the Walker environment.
>
> # Other Weaknesses - Running Time
> We would like to clarify that the reported planning frequencies were recorded on dated hardware (GTX 2080 TI). On modern hardware (RTX 4090) the planning frequencies are at least 2-4Hz depending on the problem size. We will update these numbers in the manuscript. Further, we expect that the inference time could easily be doubled with a more efficient implementation of the Sinkhorn Algorithm using JIT-compilation [7] or writing specialized kernels [8].
>
> # Question on Expert mismatch
> As our expert demonstrations may be rough and partial, our method can imitate expert demonstrations sourced from a slightly different embodiment (e.g. the new evaluations on Walker reuse the expert demonstrations for Cheetah).
>
> ---
> We hope this addresses your questions, and we are happy to elaborate on points that were inadequately covered due to character limitations.
>
> References:
> 1. Pirotta et. al. Fast Imitation via Behavior Foundation Models. ICLR ‘24
> 2. Pathak et. al. Zero Shot Visual Imitation. ICLR ‘18
> 3. Touati et. al. Learning one Representation to optimize all Rewards. NeurIPS ‘21
> 4. Genevay et. al. Sample Complexity of Sinkhorn Convergences. AISTATS ‘19
> 5. Weed et al. Sharp Asymptotic and Finite-Sample Rates of Convergence of Empirical Measures in Wasserstein Distance.
> 6. Andrychowicz et. al. Hindsight Experience Replay. NeurIPS ‘17
> 7. Cuturi et. al. Optimal Transport Tools (OTT) arXiv:2201.12324
> 8. Feydy. et. al. Interpolating between Optimal Transport and MMD using Sinkhorn Divergences. AISTATS ‘19

---

> > ### Comment · Reviewer_H8vF · 2025-04-02
> >
> > Thanks for the detailed rebuttal; I think it addresses most of my concerns, especially the major ones such as the definition of zero-shot and world model. I will now increase my score from 2 to 3, though I still would like to point out that the planning frequency still seems quite slow for real-world tasks and is a limitation of the proposed method given the authors' response.

---

> > > ### Author Response · Authors · 2025-04-08
> > >
> > > We sincerely thank the reviewer for raising their score.
> > >
> > > We agree that ZILOT’s current inference speed represents a limitation. However, current trends suggest that this limitation may be resolved.
> > > First, as noted in our earlier response, more efficient implementations of the Sinkhorn algorithm have been developed and could potentially offer up to a 2× speedup.
> > > Second, the performance improvements we observed when transitioning from older to newer hardware suggest that further 2× gains in inference speed are likely over the next 1–2 years as hardware continues to advance.
> > > Finally, policies operating at 2–10Hz are already effective as high-level controllers in robotics applications. For instance, both OpenVLA [1] and Diffusion Policy [2] operate within this range.
> > >
> > > [1] Kim et. al., OpenVLA: An Open-Source Vision-Language-Action Model. CoRL ‘24
> > >
> > > [2] Cheng et. al., Diffusion Policy: Visuomotor Policy Learning via Action Diffusion. RSS ‘23

---

### Official Review · Reviewer_LRC1 · 2025-03-11

**Overall Recommendation:** 2

**Summary:**

- The paper concerns zero-shot offline model-based imitation learning

 - A new method is proposed based on Optimal Transport to match the occupancy measure of the learning and expert policies

 - Experiments conducted on three tasks (Fetch, HalfCheetah, PointMaze), comparing the proposed method with existing baselines

**Claims And Evidence:**

- The main claim is that the proposed method optimizes an occupancy matching objective using Optimal Transport. In terms of methodological contribution, the approach appears novel. Using Optimal Transport for occupancy matching is a sound idea.

- The paper claims to address a zero-shot imitation learning problem, but it is restricted to model-based, finite-horizon MDPs. The broader literature on imitation learning includes several efficient model-free algorithms, which limits the contribution of this work.

- The experiments are limited and insufficient to fully support the main claims. Only three tasks are considered, while MuJoCo itself contains multiple tasks, yet the authors only evaluate on HalfCheetah.

**Essential References Not Discussed:**

The paper should discuss more relevant work on model-free imitation learning and inverse reinforcement learning. These could suggest alternative approaches for handling model-free, infinite-horizon models and different ways to optimize occupancy matching.

**Experimental Designs Or Analyses:**

The experimental setup is reasonable, but as previously mentioned, the experiments themselves are weak. The authors should consider extending the study with more challenging tasks typically used in prior imitation learning research.

**Methods And Evaluation Criteria:**

- The method of using Optimal Transport for occupancy matching is reasonable.

- Besides Optimal Transport, are there alternative ways to optimize occupancy matching, such as using KL divergence? Can the authors comment on this?

- The main problem formulation is restricted to finite-horizon MDPs. Can the proposed method be extended to infinite-horizon models?

- The occupancy estimation relies on sampling, which can introduce high errors and impracticalities. Additionally, the method requires dynamics estimation, which demands significant data and introduces further estimation errors.

- The proposed method is specifically tailored to model-based settings. Can it be extended to model-free settings, which are often more practical?

- As mentioned, the experimental evaluation is **weak**—only three simple tasks are considered. Why were other MuJoCo tasks not utilized for evaluation?

**Other Comments Or Suggestions:**

Please see my questions

**Other Strengths And Weaknesses:**

- The approach is sound, and the use of Optimal Transport is an interesting direction.

 - The primary weakness is the experimental evaluation. This undermines the credibility of the claims and makes the paper not yet ready for publication.

**Questions For Authors:**

- Can the authors clarify why the proposed method is restricted to model-based settings? Could it be adapted for model-free imitation learning?

- What are the main advantages of using Optimal Transport over other occupancy matching techniques, such as KL divergence?

 - Given that the formulation applies to finite-horizon MDPs, how could this be extended to infinite-horizon settings?

- The occupancy estimation relies on sampling, which can introduce high errors. Have the authors considered methods to reduce estimation errors?

 - How does the method perform in more complex tasks beyond the three considered in the experiments? Why were additional MuJoCo tasks not included?

- How does this work compare to prior model-free approaches in imitation learning and inverse reinforcement learning?

 - Could the authors discuss the scalability of their approach when applied to larger or more complex domains?

**Relation To Broader Scientific Literature:**

The contributions relate to reinforcement learning, imitation learning, zero-shot learning, and optimal transport. The approach has potential applications in fields where imitation learning is valuable, such as healthcare and autonomous driving, particularly when datasets are limited

**Theoretical Claims:**

The paper provides minimal theoretical claims and proofs.

---

> ### Author Rebuttal · Authors · 2025-03-31
>
> Thank you for your assessment and valuable feedback. We first address your main concern, our empirical evaluation, and then your other questions.
>
> # Q5 on further experiments
>
> Our experiments are chosen to be representative of the 3 most common types of MDPs found in robotics: manipulation (Fetch), navigation (Pointmaze), and locomotion (Cheetah). Instead of including similar environments, we focused on evaluating a diverse set of tasks in each environment (see Table 1), in order to cover the full complexity of possible behaviors.
> We have performed evaluations on additional Mujoco tasks, i.e. Walker with the same tasks as in Cheetah, and found results to be strongly consistent with Cheetah:
>
> |Task|$W_{\min}$ $\downarrow$|||GoalFraction $\uparrow$|||
> |-|-|-|-|-|-|-|
> ||Pi+Cls|MPC+Cls|ZILOT (ours)|Pi+Cls|MPC+Cls|ZILOT (ours)|
> ||||||||
> |walker-backflip|2.804±0.056|1.737±0.146|**1.273±0.205**|0.34±0.07|**0.89±0.03**|**0.92±0.06**|
> |walker-backflip-running|3.039±0.292|2.444±0.189|**1.709±0.093**|0.49±0.07|0.70±0.08|**0.81±0.09**|
> |walker-frontflip|2.688±0.400|1.830±0.185|**1.551±0.086**|0.57±0.16|**0.94±0.04**|**0.95±0.07**|
> |walker-frontflip-running|2.597±0.265|**1.937±0.172**|**1.921±0.149**|0.55±0.03|**0.63±0.16**|**0.76±0.11**|
> |walker-hop-backward|1.447±0.076|**0.872±0.032**|**0.836±0.100**|0.64±0.11|**0.78±0.06**|**0.84±0.07**|
> |walker-hop-forward|0.932±0.098|0.663±0.071|**0.467±0.044**|0.95±0.05|**0.99±0.01**|**1.00±0.01**|
> |walker-run-backward|1.290±0.148|**1.050±0.086**|**0.957±0.111**|**0.81±0.08**|**0.83±0.14**|**0.84±0.09**|
> |walker-run-forward|1.180±0.105|0.954±0.079|**0.672±0.058**|0.86±0.07|**0.97±0.03**|**0.99±0.01**|
> ||||||||
> |walker-all|1.997±0.047|1.436±0.049|**1.174±0.061**|0.65±0.03|0.84±0.02|**0.89±0.04**|
> ||||||||
>
> Our evaluation is also similar to prior work in zero-shot Imitation from offline data. [2] evaluate on different tasks in one manipulation environment. [1] uses four embodiments all of which are locomotion tasks. In comparison, our evaluation is richer.
>
> # Q1 model-free IL adaptation
>
> To the best of our knowledge, zero-shot distribution matching under OT has not been explored with model-free methods. The model-based component allows us to accurately predict and optimize finite-horizon occupancies. A model-free variant optimizing a similar OT objective could use techniques from [1], which we found to be, however, less data efficient (Section 5.2). Thus, a model-free variant represents an interesting research direction.
>
> # Q2 on OT vs. other distances
>
> The main advantages of OT over f-divergences (eg. KL) is that (1) it is more robust to empirical approximation (line 100 col 2), and (2) it can incorporate the underlying geometry of the space(s) which allows us to use an MDP-specific (learned) metric, the goal-conditioned (GC) value function in our case (line 210 col 2).
>
> # Q3 on finite horizon planning
>
> We acknowledge that our method relies on finite horizon planning in practice; as we show empirically, this is sufficient to largely avoid myopic behavior. An extension would be possible leveraging occupancy estimates (e.g., as in [1]); however, these techniques did not perform well in our evaluations (Section 5.2).
>
>
> # Q4 on sampling errors
>
> Estimation errors from sampling did not seem to impact performance in our experiments, especially as the support of the future state distribution is often modest. If the learned world model is non-deterministic, one may sample multiple trajectories from it to get a better estimate of the future state distribution.
>
> # Q6 on the comparison to IL and IRL
>
> ansewredThe large majority of methods for imitation learning and inverse RL (e.g., GAIL, IQLearn, DICE, OTR) are not zero-shot, and require demonstrations to be provided in advance, at training time. After pre-training, ZILOT is instead capable of imitating unseen trajectories. To the best of our knowledge, only two model-free approaches can achieve the same. The first approach [2] is however fundamentally myopic (Section 3), and is considered as a baseline (Cls+Policy in table 1). The second approach [1] is model-free, zero-shot, and non-myopic; however, we find it to be less data efficient, and to underperform in our evaluation (Section 5.2). Thus, ZILOT distinguishes itself from existing model-free approaches because it is zero-shot, non-myopic, and data-efficient.
>
> # Q7 on scalability
>
> ZILOT relies on a GC value function and a world model. Both of these are well-studied objects in the field, and we expect ZILOT to scale as these components improve.
>
> ---
> We hope this fully addresses your questions and comments, and we are happy to expand on any answers that had to be short due to character limitations.
>
> References:
> 1. Pirotta et. al. Fast Imitation via Behavior Foundation Models. ICLR ‘24
> 2. Pathak et. al. Zero Shot Visual Imitation. ICLR ‘18

---

### Official Review · Reviewer_8Rj6 · 2025-03-17

**Overall Recommendation:** 4

**Summary:**

The problem of greediness in goal conditioned imitation is resolved by matching goal occupancies. The proposed algorithm can learn from a single demonstration with partial observability. It is shown that minimizing wassertein distance between goal occupancies of expert and learner is equivalent to minimizing an optimal transport objective. This is utilized to learn policies. The proposed setup is compared to existing works with thorough experiments by measuring the proportion of goals achieved by each algorithm.

**Claims And Evidence:**

- ZILOT is non-myopic. Evidence theoretical from a pathological example where some other algorithms might fail, and empirically from certain environments where there are multiple goals to be achieved in a single trajectory.
- ZILOT giving good Offline and Zero-Shot performance and the approximations being enough to learn a good policy demonstrated via experiments.
- Ablation studies demonstrate the usefulness of some individual components.

**Essential References Not Discussed:**

none that I am aware of

**Experimental Designs Or Analyses:**

Did not run any code. The design description in paper and appendix seems very sound.

**Methods And Evaluation Criteria:**

Yes. Thorough experimentation. Comparison to relevant work.

**Other Comments Or Suggestions:**

- Acronym GC-RL is used without first introducing it.
- I don't understand why there is a subscript 1 in definitions of $\mathcal{D}_\beta$ and $\mathcal{D}_E$ . Line 78 col 2.

**Other Strengths And Weaknesses:**

Strengths:
- Very detailed results and diagrams showing how proposed algorithm works in comparison to others. One result in main paper, rest in appendix.

**Questions For Authors:**

none

**Relation To Broader Scientific Literature:**

Zero Shot Learning that is not Greedy.

**Theoretical Claims:**

Not in depth. No issues from surface level reading.

---

> ### Author Rebuttal · Authors · 2025-03-31
>
> We would like to thank the reviewer for their feedback and comments.
>
> Thank you for pointing out that the acronym “GC-RL” was not introduced in the paper. We will introduce it properly given a chance to update the paper.
>
> The subscripts in the definitions of $\mathcal{D}\_\beta$ and $\mathcal{D}\_E$ were meant to denote that $i$ runs from $1$ to $|\mathcal{D}\_\beta|$ and $|\mathcal{D}\_E|$ respectively.
> We will update the definitions to $\mathcal{D}\_\beta = (s\_0^i, a\_0^i, s\_1^i, a\_1^i, \dots)\_{i=1}^{|\mathcal{D}\_\beta|}$ and $\mathcal{D}\_E = (g\_0^i, g\_1^i, …)\_{i=1}^{|\mathcal{D}\_E|}$ respectively, to make this more clear.
>
> We remain available for further discussion for the rest of the rebuttal period.

---

> > ### Comment · Reviewer_8Rj6 · 2025-04-06
> >
> > Thank you for your response in the rebuttal. It clarifies things. I will keep my rating. I have noticed experimental evaluations as well as pointed out in other reviews, but I would like to keep this rating because of the theoretical contribution.

---

### Decision · Program_Chairs · 2025-05-01

**Decision:**

Accept (poster)

**Comment:**

The paper studies the setting of zero-shot imitation learning, where an imitation policy is directly computed from a single demonstration. The authors shows how translating this problem into a "greedy" goal-based problem may return poorly-performing policies and suggest ways to mitigate the issue by leveraging distance in occupancy distributions (approximated by a world model) to drive the imitation learning objective.

There is general consensus among the reviewers of the interest of the theoretical analysis revealing the potential limitations of the goal-based reduction of imitation learning. After the rebuttal, the authors have partially addressed concerns on the empirical validation by complementing the results initially available in the paper and by clarifying aspects related to the planning process. Based on this, I recommend acceptance for the paper, but I strongly encourage the authors to incorporate the comments of Reviewer LRC1. In particular, I believe that further ablations on OT vs f-divergences can be easily addressed. Furthermore, I tend to agree with the authors that a good variety of domains is already covered in the current submission, I still encourage them to assess whether expanding the empirical validation to other domains is possible.